# Towards improving saliency map interpretability using feature map smoothing

## Abstract

Input-gradient-based feature attribution methods, such as Vanilla Gradient, Integrated Gradients, and SmoothGrad, are widely used to explain image classifiers by generating saliency maps. However, these methods struggle to provide explanations that are both visually clear and quantitatively robust. Key challenges include ensuring that explanations are sparse, stable, and faithfully reflect the model's decision-making. Adversarial training, known for enhancing model robustness, have been shown to produce sparser explanations with these methods; however, this sparsity often comes at the cost of stability. In this work, we investigate the trade-off between stability and sparsity in saliency maps and propose the use of a smoothing layer during adversarial training. Through extensive experiments and evaluation, we demonstrate this smoothing technique improves the stability of saliency maps without sacrificing sparsity. Furthermore, a qualitative user study reveals that human evaluators tend to distrust explanations that are overly noisy or excessively sparse—issues commonly associated with explanations in naturally and adversarially trained models, respectively and prefer explanations produced by our proposed approach. Our findings offer a promising direction for generating reliable explanations with feature-map smoothed adversarially trained models, striking a balance between clarity and usability.

## 1 Introduction

Input gradient-based explanation methods highlight the features most influential to a model's decision by calculating the gradient of the model's output with respect to its input, visualized as saliency maps in images. One of the earliest approaches, Vanilla Gradient (VG) (Simonyan et al., 2014), computes gradients across input pixels, ranking features by their gradient magnitude. While prior studies have shown that input-gradients can capture relevant information regarding a model output (Samek et al., 2016), VG suffers from noisy saliency map. Hence, various methods like Integrated Gradients (IG) (Sundararajan et al., 2017), and SmoothGrad (SG) (Smilkov et al., 2017) have been proposed that modifies the input-gradient approach to reduce saliency map noise and improve the visual quality of the explanations.

However, quality explanations require more than visual appeal. Explanations should be comprehensible to users and satisfy quantitative measures to ensure their practical utility. Key properties include sparsity, which ensures explanations focus on the most relevant features by discarding irrelevant ones (Chalasani et al., 2020); stability, which guarantees consistent explanations across small input perturbations (Alvarez-Melis & Jaakkola, 2018); and faithfulness, ensuring that the explanations accurately reflect the model's actual decision-making process (Rong et al., 2022). These attributes are essential for explanations to be trustworthy and actionable in real-world applications.

In this work, we take a complementary approach by studying the quality of saliency maps in naturally and adversarially trained models and demonstrate a way to enhance above-mentioned properties of explanations in input-gradient based methods. We consider three representative input-gradient based methods (Vanilla Gradient (VG) (Simonyan et al., 2014), Integrated Gradients (IG) (Sundararajan et al., 2017), and Smooth-Grad (SG) (Smilkov et al., 2017) ) and first demonstrate that the stability of their explanations is closely tied to the model's sensitivity to input perturbations. Adversarial training (Goodfellow et al., 2015), a technique

commonly used to improve model robustness, results in explanations that are sparser, aligning with previous studies (Chalasani et al., 2020; Etmann et al., 2019). However, we observe that this increased sparsity comes at the cost of reduced stability in explanations. To mitigate this trade-off, we introduce a smoothing layer applied during adversarial training. Our extensive experiments with FMNIST, CIFAR-10 and ImageNette demonstrate that including feature-map smoothing using local filters like mean, median or Gaussian during adversarial training preserves stability of explanations without sacrificing on sparsity. Additionally, we observe that these smoothing strategies improve the faithfulness of VG and IG explanations on CIFAR-10 and ImageNette. However, such improvements are not observed for FMNIST with VG and IG, nor for SG across all three datasets.

*In addition*, we conduct a qualitative study to assess the comprehensibility of these explanations in human subjects. We interview 65 graduate students specializing in computer vision to assess their understanding of different types of explanations, which varies in terms of sparsity and smoothness. We use the Hoffman satisfaction scale as our assessment tool (Hoffman et al., 2023). Our findings reveal that explanations of input-gradient based attribution methods in naturally trained models are perceived as noisy and untrustworthy, while highly sparse explanations in adversarially trained models are also problematic due to the loss of information for enhancing sparsity. Explanations generated by input-gradient attribution methods for feature-map smoothed models are rated as more comprehensible, striking a balance between sparsity and clarity.

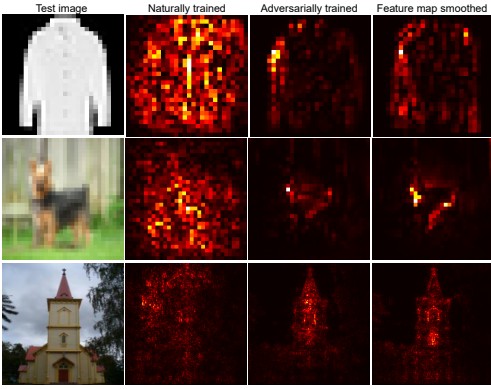

Figure 1: Saliency maps examples using Vanilla Gradient for different models that correctly classify the test images. Natural models produce noisy saliency maps ($2^{nd}$ column), adversarial models produce sparser maps ($3^{rd}$ column), and feature-map smoothed models smoothens the sparse maps ($4^{th}$ column), improving comprehensibility.

Figure 1 shows examples of saliency maps for different models on FMNIST, CIFAR-10, and ImageNette test images using Vanilla Gradient. We observe that saliency maps (a) for naturally trained models (second column) are noisy, and difficult to comprehend, (b) for adversarially trained models (third column) are sparse and align with the contours of the input image, but overly sparse saliency maps can lead to incomplete model understanding, and (c) for adversarially trained models with feature-map smoothing (fourth column) shows a reduction in sparsity to strike a balance between clarity and comprehensiveness. The smoothing helps reduce noise in the saliency map, resulting in explanations that are more continuous and coherent, while still maintaining a focus on key regions. Visualizations for Integrated Gradients, SmoothGrad and additional visualizations for Vanilla Gradient are provided in the Appendix J.

## 2 Related work

As highlighted by Ilyas et al. (2019), explanations that are meaningful and faithful to the model's decision-making process cannot be pursued independently from the training of the model, a principle central to our approach. Below we discuss such related works.

**Improving saliency maps by training modification**: Previous studies have proposed several modifications to model training to improve saliency maps. For instance, Kim et al. (2019) introduce layer-wise thresholding during backpropagation, while Dombrowski et al. (2019) suggest soft-plus activations as an alternative to ReLU for refining saliency maps. Wicker et al. (2023) develop a framework for certifying the robustness of explanations through training constraints. Meanwhile, Chenyang & Chan (2023) propose training object detectors by ensuring explanation consistency within same object and distinctions between different objects. In contrast, we do not make such modifications, and enhance the quality of explanations by applying simple smoothing filters during adversarial training.

**Study of saliency maps in adversarially trained models**: Some previous works have also explored saliency map quality in robust models (Etmann et al., 2019; Zhang & Zhu, 2019; Chalasani et al., 2020; Mangla et al., 2020; Shah et al., 2021), typically evaluating sparsity, or visual quality. Chalasani et al. (2020) show that adversarial training with $L_\infty$ attacks leads to sparse saliency maps, and theoretically demonstrate that training a 1-layer network by encouraging stability of explanations is equivalent to adversarial training, but do not present results on multi-layer networks. Etmann et al. (2019) explain the interpretability of robust models by demonstrating alignment between image and saliency maps, which works well for smaller datasets like MNIST but does not scale to larger datasets like ImageNet. Zhang & Zhu (2019) argue that adversarially trained models produce shape-biased representations, resulting in sparser saliency maps. In contrast, we approach the quality of saliency maps via the stability of the input-gradient explanation methods and establish a theoretical connection with model sensitivity, and propose adversarial training with feature-map smoothing as the mitigation of sparsity-stability tradeoff.

## 3 Method

**Preliminaries:** Consider a differentiable function $F(\mathbf{x})$, which represents a deep neural network. For simplicity, let us examine a single-layer model with the form $F(\mathbf{x}) = H(\langle \mathbf{w}, \mathbf{x} \rangle)$, where $H$ is a differentiable scalar-valued activation function (e.g., sigmoid), $\langle \mathbf{w}, \mathbf{x} \rangle$ is the dot product between the weight vector $\mathbf{w}$ and input $\mathbf{x} \in \mathbb{R}^d$. The Vanilla Gradient (VG) method (Simonyan et al., 2014) measures the sensitivity of the model output $F(\mathbf{x})$ with respect to each feature of the input $\mathbf{x}$. This is given by computing the gradient of the output $F(\mathbf{x})$ with respect to the input $\mathbf{x}$. The Integrated Gradients (IG) method (Sundararajan et al., 2017) averages the gradients along a straight-line path from a baseline input $\mathbf{x}'$ (often a zero vector) to the actual input $\mathbf{x}$. SmoothGrad (SG) (Smilkov et al., 2017) improves on any gradient-based explanations like VG or IG by adding random noise to the input $\mathbf{x}$ multiple times, calculating the explanations for each noisy version, and then averaging the results. While these methods are widely used, their stability—how consistent the explanations remain under small perturbations—is crucial for reliability. We next establish a formal connection between model sensitivity and explanation stability.

### 3.1 Relationship between explanation stability and model sensitivity

We first compute explanation using VG given by:

$$VG(\mathbf{x}) = \frac{\partial F(\mathbf{x})}{\partial \mathbf{x}} = \frac{\partial H(\langle \mathbf{w}, \mathbf{x} \rangle)}{\partial \mathbf{x}} = H'(\langle \mathbf{w}, \mathbf{x} \rangle)\mathbf{w} \tag{1}$$

Here, $H'(\langle \mathbf{w}, \mathbf{x} \rangle)$ is the gradient of activation function $H$ with respect to the $\langle \mathbf{w}, \mathbf{x} \rangle$. For example, for a sigmoid activation function, $H'(z) = H(z)(1 - H(z))$ where $z = \langle \mathbf{w}, \mathbf{x} \rangle$. This gives the VG attribution as:

$$VG^F(\mathbf{x}) = H(\langle \mathbf{w}, \mathbf{x} \rangle)(1 - H(\langle \mathbf{w}, \mathbf{x} \rangle))\mathbf{w} = F(\mathbf{x})(1 - F(\mathbf{x}))\mathbf{w} \tag{2}$$

Similarly, the IG feature attribution score for feature $i$ of input image $\mathbf{x} \in R^d$ with baseline $\mathbf{u}$ for model $F$ is given by Eqn. 3:

$$IG_i^F(\mathbf{x}, \mathbf{u}) = (x_i - u_i). \int_{\alpha=0}^{1} \partial_i F(\mathbf{u} + \alpha(\mathbf{x} - \mathbf{u}))\partial\alpha \tag{3}$$

Using a closed-form expression from Chalasani et al. (2020), IG can be rewritten as Eqn. 4:

$$IG^F(\mathbf{x}, \mathbf{u}) = [F(\mathbf{x}) - F(\mathbf{u})]\frac{(\mathbf{x} - \mathbf{u}) \odot \mathbf{w}}{\langle \mathbf{x} - \mathbf{u}, \mathbf{w} \rangle} \tag{4}$$

For SG, we add Gaussian noise $\mathbf{n} \sim \mathcal{N}(0, \sigma^2)$ to the input $\mathbf{x}$ and compute the input-gradient for multiple noisy samples $\mathbf{x}_k = \mathbf{x} + \mathbf{n}_k$ for $k = 1, \ldots, N$, where $N$ is the number of noise samples. SG explanation, when aggregating VG, is given by:

$$SG(\mathbf{x}) = \frac{1}{N}\sum_{k=1}^{N}\frac{\partial F(\mathbf{x}_k)}{\partial \mathbf{x}_k} = \frac{1}{N}\sum_{k=1}^{N}\frac{\partial H(\langle \mathbf{w}, \mathbf{x}_k\rangle)}{\partial \mathbf{x}_k} = \frac{1}{N}\sum_{k=1}^{N}H'(\langle \mathbf{w}, \mathbf{x}_k\rangle).\mathbf{w} = \frac{1}{N}\sum_{k=1}^{N}F(\mathbf{x_k})(1 - F(\mathbf{x_k}))\mathbf{w} \quad (5)$$

Now consider $\mathbf{x}' \in \mathcal{N}_{\mathbf{x}}$ is a noisy version of input image $\mathbf{x}$ where $\mathcal{N}_{\mathbf{x}}$ indicates a neighborhood of inputs $\mathbf{x}$ where the model prediction is locally consistent. The stability of explanations-VG, IG and SG-can be computed by measuring the norm of the difference between the original explanation and explanation for the noisy image. Using Eqns. 1, 4 and 5, we obtain,

$$\Delta_{VG} = ||VG^F(\mathbf{x}') - VG^F(\mathbf{x})||_1 \leq (F(\mathbf{x}') - F(\mathbf{x})).\mathbf{w} \quad (6)$$

$$\Delta_{IG} = ||IG^F(\mathbf{x}', \mathbf{u}) - IG^F(\mathbf{x}, \mathbf{u})||_1 \approx ||IG^F(\mathbf{x}', \mathbf{x})||_1 \approx \left|\left|[F(\mathbf{x}') - F(\mathbf{x})]\frac{(\mathbf{x}' - \mathbf{x}) \odot \mathbf{w}}{\langle \mathbf{x}' - \mathbf{x}, \mathbf{w}\rangle}\right|\right|_1 \quad (7)$$

$$\Delta_{SG} = \sum_{k=1}^{N}||SG^F(\mathbf{x}') - SG^F(\mathbf{x})||_1 \leq \frac{1}{N}(F(\mathbf{x}') - F(\mathbf{x})).\mathbf{w} \quad (8)$$

Since $\mathbf{w}$ is fixed for a given model, the bounds in Eqns 6, 7 and 8 indicate that the stability of explanations is influenced by the model sensitivity $F(\mathbf{x}') - F(\mathbf{x})$, setting up a basis for using methods that enhance explanation stability by reducing model sensitivity. However, these bounds do not serve a strict proportional relationship between model sensitivity and attribution stability, and should not be interpreted as such. Rather, the bounds serve as approximate indicators, highlighting that attribution stability is influenced by model sensitivity. For a detailed derivation, see Appendix H, and for conditions affecting the tightness of these bounds, refer to Appendix F.

### 3.2 Adversarial training and impact on saliency map stability

Building on the observations from Section 3.1, various regularization strategies can enhance the quality and stability of saliency maps. One such approach is natural training-based regularization, which involves incorporating explicit smoothness constraints on the model's gradients or feature activations. A fundamental technique in this category is input noise regularization, where Gaussian noise is injected into training samples during optimization (Bishop, 1995). This method has been shown to produce smoother and more reliable saliency maps (Smilkov et al., 2017).

To explicitly address the issue of model sensitivity to worst-case perturbations, we consider adversarial training (Goodfellow et al., 2015) as a method for improving robustness. Adversarial training modifies the loss function to minimize sensitivity to input perturbations by solving $\mathbb{E}_{(\mathbf{x},y)\sim D}\left[\max_{||\delta||_\infty \leq \epsilon} \mathbb{L}(\mathbf{x} + \delta, y; \mathbf{w})\right]$ where $\delta$ is a small, worst-case perturbation and $\epsilon$ is the perturbation bound. The inner maximization finds the most adversarial perturbation within the constraint $||\delta||_\infty \leq \epsilon$, while the outer minimization ensures that the model learns to be invariant to such perturbations.

In Figure 3, given a test image from the ImageNette dataset, we visualize feature maps derived from (a) a naturally trained model and (b) an adversarially trained model. All models use the identical ResNet18 architecture (He et al., 2016) and training settings. Feature maps are extracted from the first residual block (which consists of 128 channels), and three representative channels are shown for comparison. A key observation is that adversarial training shrinks many feature activations, leading to more selective attention in learned representations. This behavior directly affects input-gradient-based saliency maps, making them sparser in adversarially trained models compared to naturally trained ones (see Figure 3(d)). This effect is also explored by Etmann et al. (2019); Chalasani et al. (2020).

However, adversarial training does not necessarily improve explanation stability in deep networks. As we demonstrate in Sections 4.1 and 4.3, while adversarial training enforces sparsity in saliency maps, it does not guarantee stability and comprehensibility of saliency maps. This leads to a trade-off: *sparser explanations may enhance readability but can also reduce attribution stability*. Our findings suggest that while adversarial

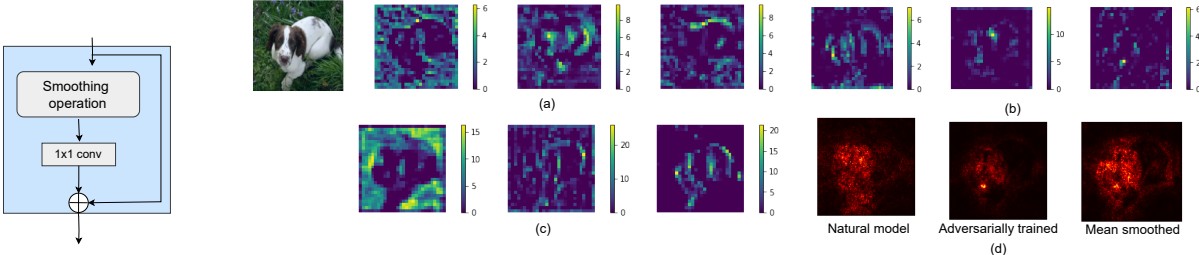

Figure 2: Feature-map smoothing block

Figure 3: Plot of feature maps (channel=7, 21, 127) after first residual block for a test image on different ResNet18 ImageNette models: (a) a naturally trained model, (b) an adversarially-trained model, (c) an adversarially-trained model with feature-map smoothing (mean filter) (d) corresponding saliency maps using Vanilla Gradient.

training enhances sparsity of saliency maps, additional mechanisms (such as feature-map smoothing) may be required to preserve the stability of gradient-based explanations.

### 3.3 Feature map smoothing for comprehensible explanations

To address the limitations of adversarial training on saliency map stability and comprehensibility, we incorporate feature map smoothing (Xie et al., 2019). By smoothing out the sharp reductions in feature activations, these smoothing techniques help stabilize input-gradient-based explanations, producing saliency maps that are both sparse and stable, when combined with adversarial training.

Feature map smoothing can also be considered as a regularization technique that regularizes intermediate feature activations, preventing sudden activation changes that negatively impact interpretability. This is particularly relevant for gradient-based explanations, as smoother activations lead to more structured and comprehensible saliency maps. Unlike direct input perturbation methods such as SmoothGrad (Smilkov et al., 2017), which focus on reducing variance in saliency maps by averaging gradients across noisy samples, feature map smoothing operates at the representation level by enforcing local consistency in feature activations.

In our study, we explore three local-smoothing filters (mean, median, and Gaussian) and two non-local smoothing filters (non-local Gaussian and embedded Gaussian) (Wang et al., 2018), leveraging their complementary properties in stabilizing feature maps. Local smoothing filters reduce noise by averaging feature activations in a small spatial neighborhood. For example, a mean filter replaces each feature with the average of nearby features within a defined kernel. A median filter, unlike a mean filter, computes the median value within a small sliding window over the feature map. A Gaussian filter applies a smoothing effect to feature maps by convolving them with a Gaussian kernel, effectively reducing Gaussian noise. Unlike local-smoothing filters, non-local filters consider long-range dependencies, preserving important structural patterns in activations. Given a feature map $x$, such filters calculate a weighted average of features across all spatial positions within the set $\mathcal{L}$, given by $m_i = \frac{1}{\mathcal{C}(x)} \sum_{\forall j \in \mathcal{L}} f(x_i, x_j).x_j$ where $f(x_i, x_j)$ is feature dependent weighting function and $\mathcal{C}(x)$ is a normalization function. Non-local gaussian filter formulates the feature dependent function as the dot-product similarity between the feature maps $f(x_i, x_j) = e^{(x_i^T x_j)}$. Embedded Gaussian computes similarity in embedding space by computing embedded versions of the feature map $x$, given by $f(x_i, x_j) = e^{(\theta(x_i)^T \eta(x_j))}$ where, $\theta(x_i) = W_\theta x_i$ and $\eta(x_j) = W_\phi x_j$ are the embeddings of feature maps, obtained after $1 \times 1$ convolution. See Appendix B for more discussion on each filter.

Figure 2 illustrates a feature-map smoothing block, which can be applied to any feature map within a model. This block consists of a smoothing operation, followed by a 1x1 convolutional layer. The feature map is then merged with the original input via a residual connection, ensuring that the model retains essential feature information while benefiting from the smoothing operation. The introduction of this smoothing block has minimal impact on model accuracy (see Appendix C), yet it significantly alters the behavior of gradient-based explanations, leading to more stable and interpretable saliency maps.

As shown in Figure 3(c), applying feature map smoothing to an adversarially trained model introduces a noticeable smoothing effect, which varies depending on the type of filter used. For example, mean filtering reduces rapid fluctuations in feature map values by averaging neighboring activations. While adversarial training alone (Figure 3(b)) shrinks feature activations leading to discontinuities in the learned representations, the addition of smoothing alleviates this issue by preserving key feature activations while eliminating high-frequency artifacts typically seen in naturally trained models. This results in smoother and more interpretable saliency maps, as illustrated in Figure 3(d). Furthermore, feature map smoothing aligns with the theoretical stability bounds derived in Section 3.1. Since smoothing reduces the norm of feature variations—specifically, $\|F(\mathbf{x}') - F(\mathbf{x})\|$—this leads to tighter stability bounds for Vanilla Gradients (VG), Integrated Gradients (IG), and SmoothGrad (SG).

In Appendix G, we also analyze the effect of the convolution operation on receptive field expansion within the smoothing block. We modify the feature smoothing block so that it performs only a convolution (identity or randomly initialized) operation. This modified setup ensures that there is only an expansion of the receptive field without filtering operations and it can provide a baseline study to analyze the effect of receptive field expansion on its own. Our experiments show that feature map smoothing provides a competitive advantage, in terms of sparsity and stability of saliency maps. By effectively balancing robustness, interpretability, and stability, adversarial training with feature-map smoothing enhances the utility of saliency maps in real-world applications.

# 4 Experiment and Analysis

## 4.1 Experiment Framework

**Setup:** We evaluate our approach on three datasets: FMNIST (Xiao et al., 2017), CIFAR-10 (Krizhevsky et al., 2009), and ImageNette (Howard, 2020), training several model variants for each. The variants include: 1) naturally trained (N), 2) adversarially trained (A), 3) adversarial training with mean-filter smoothing (M1), 4) adversarial training with median-filter smoothing (M2), 5) adversarial training with Gaussian-filter smoothing (G), 6) adversarial training with embedded filter smoothing (E), and 7) adversarial training with non-local Gaussian smoothing (NG). Following the setup from Chalasani et al. (2020), we use LeNet (LeCun et al., 1998) for FMNIST and Wide-ResNet (Zagoruyko & Komodakis, 2016) for CIFAR-10. We use ResNet-18 He et al. (2016)for ImageNette. For adversarial training, we apply perturbations under the $L_\infty$ norm using the PGD attack (Madry et al., 2018). The models are trained with $\epsilon = 0.1$ for FMNIST and CIFAR-10, and $\epsilon = 1/255$ for ImageNette, as these values yielded the best performance across our evaluations. We achieved optimal results by adding the smoothing block after the first convolutional or residual block. We discuss the impact of altering the smoothing block's position in Appendix D. Full details of our datasets and training methodology are provided in Appendix A. We also discuss the effect of feature map smoothing on saliency map quality for a different network architecture in Appendix E.

**Evaluation Metrics:** Given a saliency map from Vanilla Gradient (VG), Integrated Gradients (IG) and SmoothGrad (SG) for each model and dataset, we compute its sparseness using Gini index (G) (Chalasani et al., 2020), and its stability using relative input stability (RIS), relative output stability (ROS) and relative representation stability (RRS) (Agarwal et al., 2022). We analyze faithfulness using ROAD analysis (Rong et al., 2022) and saliency map similarity using structural similarity index (SSIM) (Adebayo et al., 2018). In addition, we quantify the ROAD plot using area under perturbation curve, computed as $ROAD_{AUC} = \frac{1}{L+1} \sum_{k=1}^{L} \langle f(x^{(0)}) - f(x^{(k)}) \rangle$ where, $L$ represents the number of feature removal steps, and $f(x)$ is the classifier's output probability for the originally predicted class given the input $x$. The term $x^{(0)}$ corresponds to the unperturbed input image, while $x^{(k)}$ represents the image after $k$ perturbation steps. All results are aggregated for 1000 randomly selected test images that the model accurately classifies across all datasets. See Appendix I for detail discussion on metrics. Our code is available at `https://anonymous.4open.science/r/iclr2025xai/README.md`.

## 4.2 Results and discussion

Similar to Chalasani et al. (2020), we compare the sparsity, stability and faithfulness improvement of saliency maps with respect to the naturally trained model (N). Specifically, for a given training method (M), we compute the following metrics that quantify the improvement in sparseness (dG), relative input stability (dRIS), relative output stability (dROS), relative representation stability (dRRS) and faithfulness (dROAD) of the explanation method $\phi(.) \in \{VG, IG, SG\}$

$$dG[\phi(\mathbf{x})] = G^M[\phi(\mathbf{x})] - G^N[\phi(\mathbf{x})]$$
$$dRIS[\phi(\mathbf{x})] = RIS^M[\phi(\mathbf{x})] - RIS^N[\phi(\mathbf{x})]$$
$$dROS[\phi(\mathbf{x})] = ROS^M[\phi(\mathbf{x})] - ROS^N[\phi(\mathbf{x})]$$
$$dRRS[\phi(\mathbf{x})] = RRS^M[\phi(\mathbf{x})] - RRS^N[\phi(\mathbf{x})$$
$$dROAD[\phi(\mathbf{x})] = ROAD_{AUC}^M[\phi(\mathbf{x})] - ROAD_{AUC}^N[\phi(\mathbf{x})]$$

### 4.2.1 On the sparsity and stability of saliency maps

Table 1: Sparsity-Stability-Faithfulness evaluation of Vanilla Gradient (VG), Integrated Gradients (IG) and SmoothGrad (SG) on the following FMNIST, CIFAR-10 and ImageNette models: naturally trained (N), adversarially trained (A), adversarial training with mean-filter smoothing (M1), adversarial training with median-filter smoothing (M2), adversarial training with Gaussian-filter smoothing (G), adversarial training with embedded filter smoothing (E), and adversarial training with non-local Gaussian smoothing (NG). ↑ and ↓ indicate that larger & smaller values are better respectively. Results show that increase in sparsity comes at the expense of stability.

| | | FMNIST | | | | | | CIFAR-10 | | | | | | ImageNette | | | | | |
| --- | --- | --- | --- | --- | --- | --- | --- | --- | --- | --- | --- | --- | --- | --- | --- | --- | --- | --- | --- |
| | | **A** | **M1** | **M2** | **G** | **E** | **NG** | **A** | **M1** | **M2** | **G** | **E** | **NG** | **A** | **M1** | **M2** | **G** | **E** | **NG** |
| **VG** | dG ↑ | 0.198 | 0.198 | 0.171 | 0.183 | 0.188 | **0.219** | 0.188 | 0.185 | 0.181 | 0.185 | 0.189 | **0.190** | 0.050 | 0.018 | 0.036 | 0.063 | **0.117** | 0.107 |
| | dRIS ↓ | 2.193 | 1.396 | **-1.025** | 1.168 | -0.400 | 1.781 | -0.458 | -0.621 | **-0.676** | -0.465 | -0.503 | -0.637 | -0.056 | **-0.121** | -0.016 | -0.098 | 0.767 | 0.401 |
| | dROS ↓ | 2.084 | 1.121 | **-1.222** | 0.739 | -0.451 | 1.785 | 0.217 | 0.260 | **0.214** | 0.226 | 0.280 | 0.257 | -0.362 | **-0.470** | -0.297 | -0.456 | 0.386 | 0.240 |
| | dRRS ↓ | 2.489 | 1.600 | **-0.799** | 1.452 | -0.126 | 2.202 | 0.445 | 0.453 | **0.433** | 0.457 | 0.438 | 0.467 | 0.241 | **-0.218** | -0.096 | -0.078 | 0.778 | 0.441 |
| | dROAD ↑ | -0.005 | -0.045 | -0.050 | **0.006** | 0.001 | -0.018 | 0.029 | 0.036 | 0.035 | 0.039 | 0.034 | **0.040** | -0.008 | **0.151** | 0.014 | 0.053 | 0.095 | 0.004 |
| **IG** | dG ↑ | 0.067 | 0.075 | 0.047 | 0.050 | 0.021 | **0.069** | 0.091 | 0.091 | 0.092 | 0.094 | 0.087 | **0.095** | 0.034 | 0.033 | 0.062 | 0.041 | **0.063** | 0.056 |
| | dRIS ↓ | 2.016 | 2.679 | **-0.843** | 4.564 | 1.007 | 2.714 | -1.056 | -1.504 | **-1.862** | -1.662 | -1.499 | -1.597 | 0.143 | **-0.0071** | 0.135 | 0.276 | 0.370 | 0.163 |
| | dROS ↓ | 1.931 | 2.917 | **-0.698** | 4.681 | 2.103 | 2.526 | 0.228 | 0.350 | **-0.123** | -0.090 | 0.593 | 0.041 | -0.230 | **-0.532** | -0.451 | -0.376 | -0.273 | -0.038 |
| | dRRS ↓ | 2.037 | 2.811 | **-0.741** | 5.030 | 1.622 | 2.676 | 1.050 | 0.219 | **0.163** | 0.410 | 0.258 | 0.243 | -0.157 | -0.121 | -0.027 | **-0.224** | 0.135 | -0.232 |
| | dROAD ↑ | -0.010 | -0.070 | -0.029 | -0.035 | **0.049** | -0.034 | 0.071 | 0.076 | 0.092 | 0.069 | 0.088 | **0.098** | 0.064 | 0.045 | 0.017 | 0.051 | **0.066** | 0.058 |
| **SG** | dG ↑ | 0.198 | 0.198 | 0.171 | 0.183 | 0.158 | **0.219** | 0.681 | 0.684 | 0.684 | 0.678 | 0.684 | **0.686** | 0.036 | 0.028 | 0.064 | 0.035 | **0.101** | 0.068 |
| | dRIS ↓ | 0.945 | 0.799 | **-0.466** | 0.994 | -0.282 | 2.015 | -0.040 | -0.034 | **-0.191** | 0.885 | 0.372 | 0.340 | 0.017 | **-0.148** | 0.719 | 0.045 | 0.272 | 0.030 |
| | dROS ↓ | 5.593 | 3.418 | **-0.194** | 2.034 | 1.099 | 2.988 | 4.619 | 5.087 | **4.393** | 4.540 | 4.733 | 0.494 | -0.576 | **-0.728** | -0.589 | -0.657 | -0.331 | -0.348 |
| | dRRS ↓ | -1.360 | 0.028 | -0.850 | -0.694 | **-2.085** | 1.245 | -2.561 | -2.469 | **-2.693** | -2.612 | -2.440 | -2.582 | -0.274 | **-0.381** | -0.216 | -0.306 | 0.010 | -0.234 |
| | dROAD ↑ | 0.006 | 0.000 | -0.018 | -0.009 | **0.262** | -0.016 | 0.003 | -0.004 | 0.012 | 0.030 | **0.127** | 0.033 | **0.021** | 0.000 | 0.019 | -0.004 | -0.008 | 0.001 |

As illustrated in Table 1, across all datasets-FMNIST, CIFAR-10 and ImageNette-all models consistently achieve positive dG values for all three explanation methods (VG, IG, SG), indicating that compared to naturally trained models, these explanation methods produce sparser saliency maps in adversarially trained models and adversarially trained models with feature map smoothing. Notably, the highest sparsity gains in explanations are observed in adversarially trained models utilizing non-local smoothing filters. On FMNIST and CIFAR-10, the NG models (adversarially trained with non-local gaussian) attain the highest sparsity across all explanation methods, and on ImageNette dataset, model E (adversarially trained with embedded gaussian) achieves the highest sparsity for Vanilla Gradient, Integrated Gradients and SmoothGrad. However, this increase in sparsity comes at the expense of stability, as most robust models exhibit reduced stability in their explanations, suggesting an inverse relationship between the sparsity and stability of saliency maps. For example: adversarially trained models with non-local gaussian filter (NG), while achieving high sparsity for explanations, show significant drops in dRIS, dROS, and dRRS, indicating that their explanations may be more sensitive to input perturbations or variations in model representations. Notably, models M1 (adversarially trained with mean filtering) and M2 (adversarially trained with median filtering) provide a promising middle-ground. On FMNIST and CIFAR-10, explanations consistently achieves the highest stability in adversarially trained models with median filtering (M2) across all methods, while still maintaining

sparsity gain. On ImageNette, adversarially trained model with mean filtering (M1) offers the best stability across explanation methods. These results suggest that the use of local smoothing filters like mean and median filters during adversarial training can preserve the stability of saliency maps while maintaining a degree of sparsity.

### 4.2.2 On the faithfulness of saliency maps

In this section, we inspect the impact of adversarial training strategy and inclusion of smoothing filters on the faithfulness of explanations. We include a random baseline (randomly sampled saliency map) for comparison of explanation faithfulness with Vanilla Gradient, Integrated Gradients and SmoothGrad.

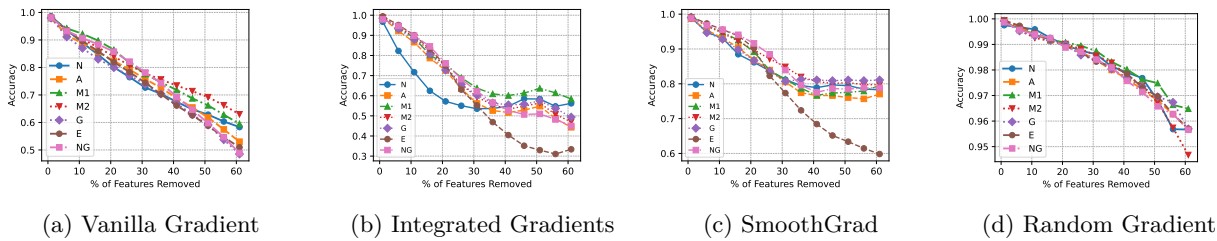

(a) Vanilla Gradient          (b) Integrated Gradients          (c) SmoothGrad          (d) Random Gradient

Figure 4: ROAD evaluation for measuring saliency map faithfulness where sharper drop in accuracy is better on various FMNIST models: naturally trained (N), adversarially trained (A), adversarial training with mean-filter smoothing (M1), adversarial training with median-filter smoothing (M2), adversarial training with Gaussian-filter smoothing (G), adversarial training with embedded filter smoothing (E), and adversarial training with non-local Gaussian smoothing (NG).

**FMNIST:** The $d$ROAD scores in Table 1 indicate that, for FMNIST, adversarial training—both with and without smoothing filters—generally leads to a reduction in faithfulness across all three explanation methods. For Vanilla Gradient (VG), all models show a drop in $d$ROAD compared to the naturally trained baseline, except for models G (adversarially trained with Gaussian filter) and E (adversarially trained with embedded filter). However, even in these two cases, the gains in faithfulness are marginal. This observation aligns with the accuracy drop trend in Figure 4a, where the differences across models are not significant. For Integrated Gradients (IG), Figure 4b shows that the naturally trained model exhibits the steepest accuracy decline when high-attribution pixels are removed, while model E (adversarially trained with embedded filter) continues to show a more gradual drop. This pattern is consistent with the $d$ROAD scores, where only model E (adversarially trained with embedded filter) achieves a positive score, indicating improved faithfulness, whereas all other variants perform worse than the naturally trained counterpart. A similar trend holds for SmoothGrad (SG): only model E (adversarially trained with embedded filter) demonstrates a meaningful gain in $d$ROAD. Finally, Figure 4d, which presents the ROAD curve for the baseline Random Gradient, shows the lowest drop in accuracy, thereby validating the reliability of the ROAD evaluation framework.

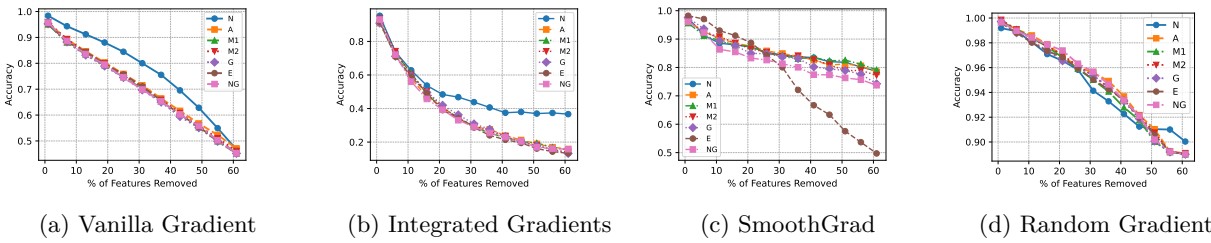

(a) Vanilla Gradient          (b) Integrated Gradients          (c) SmoothGrad          (d) Random Gradient

Figure 5: ROAD evaluation for measuring saliency map faithfulness where sharper drop in accuracy is better on various CIFAR10 models: naturally trained (N), adversarially trained (A), adversarial training with mean-filter smoothing (M1), adversarial training with median-filter smoothing (M2), adversarial training with Gaussian-filter smoothing (G), adversarial training with embedded filter smoothing (E), and adversarial training with non-local Gaussian smoothing (NG)

**CIFAR-10:** On CIFAR-10, all smoothing methods lead to improved faithfulness of Vanilla Gradient (VG) explanations, as evidenced by the positive $d$ROAD scores in Table 1. Among these, the NG model (adversarially trained with a non-local Gaussian filter) achieves the highest score. This trend is visually supported by Figure 5a, where the naturally trained model has the least steep drop in accuracy. Integrated Gradients (IG) similarly benefits from smoothing, with all variants showing improved $d$ROAD scores relative to the naturally trained baseline with the NG (adversarially trained with a non-local Gaussian filter) model achieving the highest score. This is reflected in Figure 5b, where the naturally trained model exhibits a less steep drop in accuracy compared to the other models. For SmoothGrad (SG), Table 1 shows that all models except M1 (adversarially trained with mean filter) outperform the naturally trained model in terms of $d$ROAD, with model E (adversarially trained with embedded filter) achieving the highest score. Figure 5c corroborates this result, showing the steepest accuracy drop for model E (adversarially trained with embedded filter). As in the FMNIST case, the Random Gradient baseline in Figure 5d shows minimal accuracy drop, reinforcing the validity of the ROAD evaluation metric.

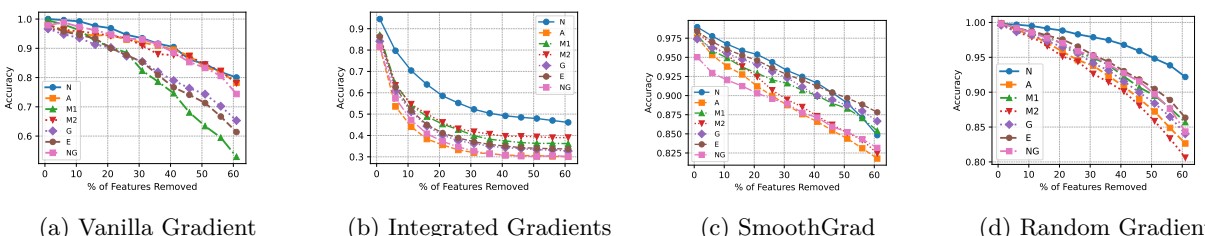

| (a) Vanilla Gradient | (b) Integrated Gradients | (c) SmoothGrad | (d) Random Gradient |

Figure 6: ROAD evaluation for measuring saliency map faithfulness where sharper drop in accuracy is better on various ImageNette models: naturally trained (N), adversarially trained (A), adversarial training with mean-filter smoothing (M1), adversarial training with median-filter smoothing (M2), adversarial training with Gaussian-filter smoothing (G), adversarial training with embedded filter smoothing (E), and adversarial training with non-local Gaussian smoothing (NG)

**ImageNette:** For Vanilla Gradient (VG), Figure 6a shows that several adversarially trained models—particularly M1 (adversarially trained with mean filter) and E (adversarially trained with embedded filter)—exhibit a more pronounced drop in accuracy compared to the naturally trained model, indicating improved faithfulness. This observation is corroborated by the $d$ROAD scores in Table 1, where M1 (adversarially trained with mean filter) and E (adversarially trained with embedded filter) achieve the highest values. In contrast, adversarial training without smoothing leads to reduced faithfulness. Integrated Gradients (IG) follows a similar trend. As shown in Figure 6b, the naturally trained model experiences a relatively flat accuracy drop, suggesting lower attribution faithfulness. Models E (adversarially trained with embedded filter) and A (adversarially trained without smoothing) achieve the steepest accuracy drops, with model NG (adversarially trained with non-local Gaussian filter) also performing competitively. These observations align with the $d$ROAD values in Table 1. For SmoothGrad (SG), both Figure 6c and the corresponding $d$ROAD scores indicate only marginal improvements across models. The best performance is observed in the adversarially trained model without smoothing, followed closely by model NG (adversarially trained with a non-local Gaussian). Overall, gains in faithfulness using SmoothGrad on ImageNette remain limited compared to VG and IG.

### 4.2.3 On the structural similarity of saliency maps

In this section, we inspect the impact of adversarial training strategy and inclusion of smoothing filters on the structural similarity of saliency maps. Following Adebayo et al. (2018), for each image $\mathbf{x}$, we introduce Gaussian noise ($\mathcal{N}(0, \sigma)$) to create its noisy counterpart $\mathbf{x}'$ while ensuring consistent model predictions. Subsequently, we compute saliency maps for $\mathbf{x}$ and $\mathbf{x}'$ and measure the structural similarity between the maps.

**FMNIST:** Figure 7a shows that as noise increases, the structural similarity index (SSIM) of Vanilla Gradient saliency maps generated from naturally trained models deteriorates rapidly. In contrast, all other models have significantly higher similarity scores. Among the smoothing techniques integrated into adversarial training,

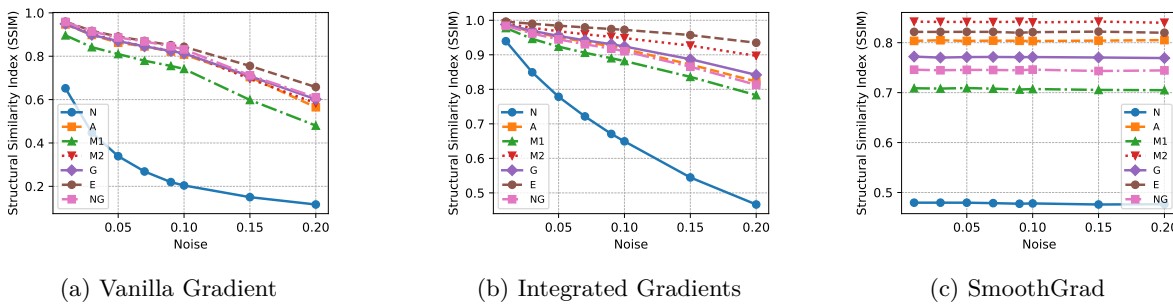

(a) Vanilla Gradient      (b) Integrated Gradients      (c) SmoothGrad

Figure 7: Structural similarity evaluation of saliency maps on various FMNIST models: naturally trained (N), adversarially trained (A), adversarial training with mean-filter smoothing (M1), adversarial training with median-filter smoothing (M2), adversarial training with Gaussian-filter smoothing (G), adversarial training with embedded filter smoothing (E), and adversarial training with non-local Gaussian smoothing (NG). Results demonstrate that naturally trained models consistently yield lower SSIM values, and adversarial training combined with smoothing methods significantly enhances structural consistency.

embedded filter smoothing (E) and non-local Gaussian smoothing (NG) demonstrates superior performance, consistently maintaining the highest SSIM across all noise levels. Figure 7b shows the structural similarity of saliency maps generated by Integrated Gradients. Similar to Vanilla Gradient, saliency maps derived from naturally trained (N) models demonstrate a rapid decrease in structural similarity (SSIM) as noise increases. In contrast, all other model significantly stabilizes these explanations, resulting in considerably higher SSIM values even at higher noise levels. Notably, adversarial training with embedded filter smoothing (E) consistently achieves the highest SSIM. Figure 7c shows the plot for SmoothGrad. Notably, all methods including naturally trained (N) show relatively constant and high SSIM values across increasing noise levels. This is primarily due to the intrinsic averaging mechanism of SmoothGrad, which already involves aggregating gradients computed over several noise-added inputs, inherently producing stable attribution maps. However, clear differences exist between the SSIM values as adversarial training enhanced by embedded (E) and median-filter smoothing (M2) consistently achieves the highest SSIM values.

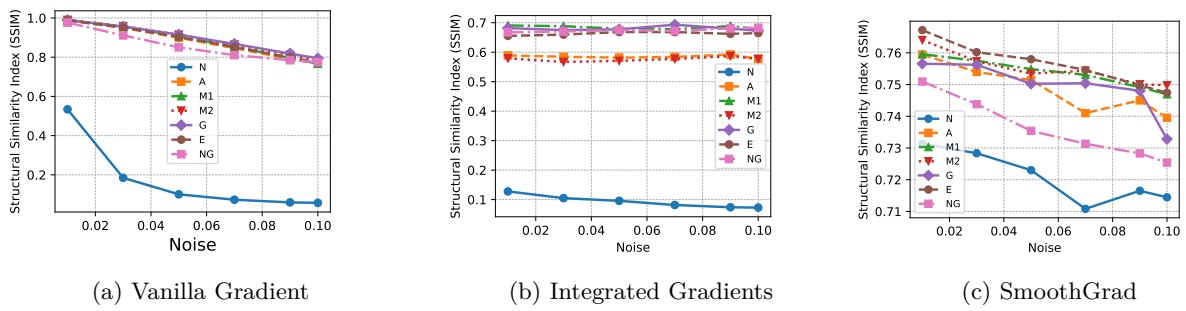

(a) Vanilla Gradient      (b) Integrated Gradients      (c) SmoothGrad

Figure 8: Structural similarity evaluation of saliency maps on various CIFAR-10 models: naturally trained (N), adversarially trained (A), adversarial training with mean-filter smoothing (M1), adversarial training with median-filter smoothing (M2), adversarial training with Gaussian-filter smoothing (G), adversarial training with embedded filter smoothing (E), and adversarial training with non-local Gaussian smoothing (NG). Results demonstrate that naturally trained models consistently yield lower SSIM values, and adversarial training combined with smoothing methods significantly enhances structural consistency.

**CIFAR-10:** The structural similarity (SSIM) of saliency maps generated by Vanilla Gradient for CIFAR-10 models is illustrated in Figure 8a, which demonstrates the rapid decline in SSIM values with increased noise in the naturally trained (N) model. In contrast, all other models show substantially improved robustness with high structural similarity even with high noise. In Figure 8b, we can observe the structural similarity (SSIM) of Integrated Gradients-based saliency maps where, naturally trained (N) models display very low structural similarity values, underscoring their susceptibility to even minor input perturbations. Conversely,

adversarially trained models with embedded (E) filter and non-local Gaussian (NG) filter consistently results in significantly higher and stable SSIM scores. Figure 8c presents the structural similarity (SSIM) of saliency maps generated using SmoothGrad. Unlike FMNIST, SmoothGrad in CIFAR-10 exhibits a subtle decrease in SSIM values with increasing noise. However, the decline in SSIM score is relatively small and SSIM score is still high, across all the models. Adversarially trained models with smoothing filters again have higher SSIM values.

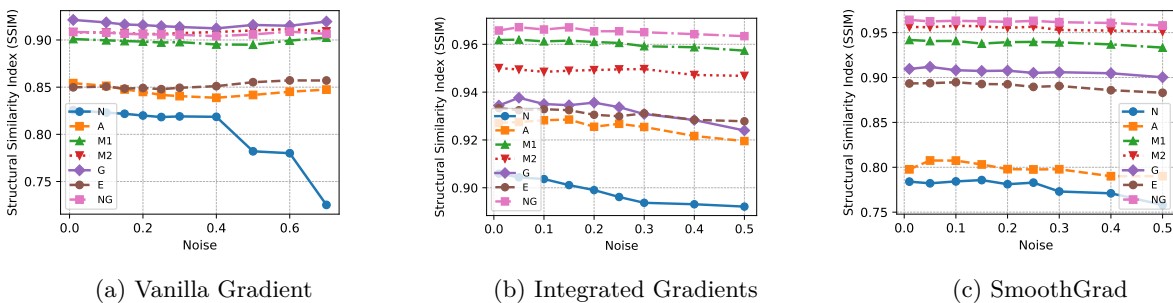

| (a) Vanilla Gradient | (b) Integrated Gradients | (c) SmoothGrad |

Figure 9: Structural similarity evaluation of saliency maps on various ImageNette models: naturally trained (N), adversarially trained (A), adversarial training with mean-filter smoothing (M1), adversarial training with median-filter smoothing (M2), adversarial training with Gaussian-filter smoothing (G), adversarial training with embedded filter smoothing (E), and adversarial training with non-local Gaussian smoothing (NG). Results demonstrate that naturally trained models consistently yield lower SSIM values, and adversarial training combined with smoothing methods significantly enhances structural consistency.

**ImageNette:** Figure 9a shows the structural similarity (SSIM) of saliency maps generated by the Vanilla Gradient explanation method for Imagenette models as the input noise increases. Again, naturally trained (N) models exhibit a high sensitivity, with SSIM significantly dropping at higher noise intensities. In contrast, all other models maintain relatively stable and notably higher SSIM values, demonstrating improved robustness, notably adversarially trained models with Gaussian-filter smoothing (G), median-filter smoothing (M2), and non-local Gaussian smoothing (NG). In Figure 9b, we illustrate the structural similarity (SSIM) of Integrated Gradients-generated saliency maps where naturally trained (N) models show the lowest SSIM values with a clear declining trend, and all other models demonstrate enhanced robustness and consistently higher structural similarity. Notably, adversarially trained models with non-local Gaussian smoothing (NG) and mean-filter smoothing (M1) achieve the highest SSIM values. Figure 9c presents the structural similarity (SSIM) of SmoothGrad-generated saliency maps. Unlike CIFAR-10 and FMNIST, Imagenette models exhibit high SSIM values with minimal variations, underscoring SmoothGrad's inherent robustness in producing stable explanations. However, adversarially trained models with non-local Gaussian smoothing (NG), mean-filter smoothing (M1), and median-filter smoothing (M2) achieve the highest and most stable SSIM scores. Naturally trained models again exhibit relatively flat SSIM values as noise increases, indicating SmoothGrad partially mitigates noise sensitivity.

### 4.2.4 Trade-off between model performance & saliency map quality

Our findings reveal that: (a) input-gradient based attribution methods produce sparse saliency maps in adversarially trained models (See Section 4.2.1), (b) adversarially trained models with non-local-feature-map smoothing, increase the sparsity of saliency maps but compromise on stability (See Table 1), (c) adversarially trained models, with local-feature-map smoothing, enhances the stability of saliency maps without compromising on sparsity (See Table 1), (d) saliency maps in adversarially trained models with feature map smoothing consistently demonstrate invariance to noise (See Section 4.2.3), and (e) saliency maps in adversarially trained models with feature map smoothing are more faithful to the underlying model than naturally trained counterparts on CIFAR-10 and ImageNette with Vanilla Gradient and Integrated Gradients (See Table 4.2.1). This aligns with the findings of Shah et al. (2021), which showed that naturally trained models fail to capture the most discriminative features, often due to feature leakage and Eberle

et al. (2022) which showed that less sparse attention vectors in transformers are less faithful to the model predictions.

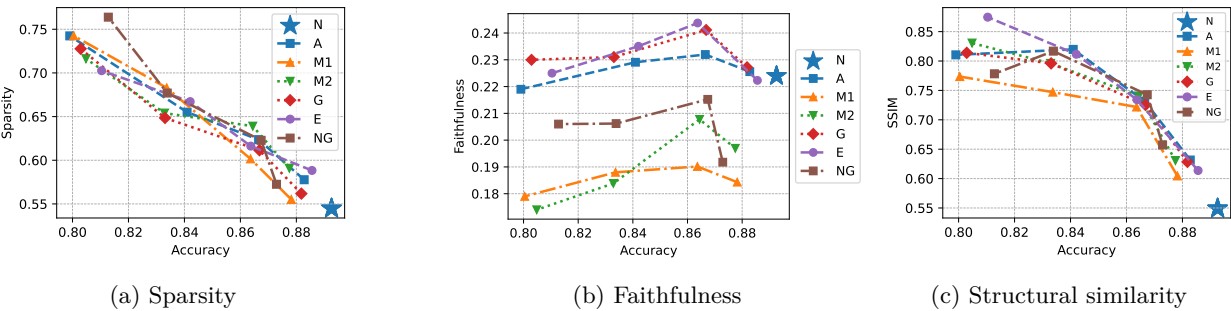

(a) Sparsity         (b) Faithfulness         (c) Structural similarity

Figure 10: Tradeoff between saliency map quality and model performance on FMNIST models : naturally trained (N), adversarially trained (A), adversarial training with mean-filter smoothing (M1), adversarial training with median-filter smoothing (M2), adversarial training with Gaussian-filter smoothing (G), adversarial training with embedded filter smoothing (E), and adversarial training with non-local Gaussian smoothing (NG). Results show that adversarially trained models (with smoothing filters) improve saliency map quality but at the expense of benign accuracy.

These observations lead to the conclusion that saliency maps in adversarially trained models (with feature map smoothing) are more reliable and interpretable than natural models for the input-gradient based attribution methods. However, it's important to note a caveat: such models come at the expense of benign accuracy.

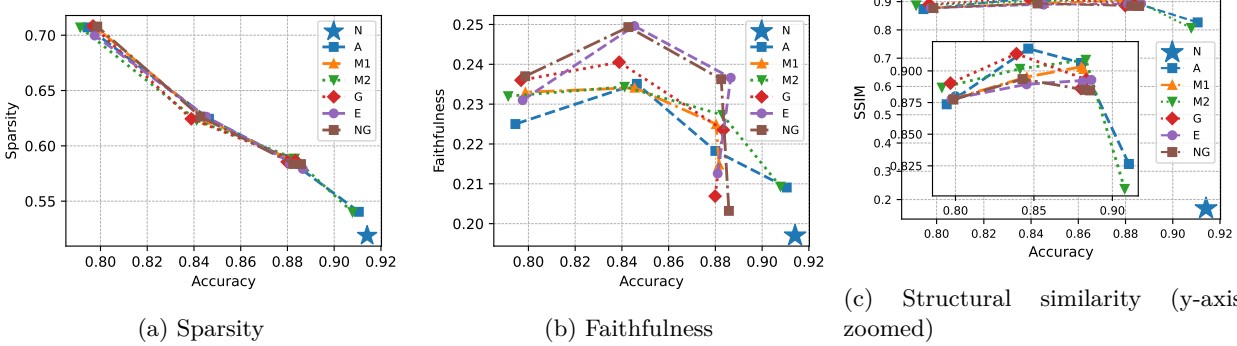

(a) Sparsity         (b) Faithfulness         (c) Structural similarity (y-axis zoomed)

Figure 11: Tradeoff between saliency map quality and model performance on CIFAR-10 models: naturally trained (N), adversarially trained (A), adversarial training with mean-filter smoothing (M1), adversarial training with median-filter smoothing (M2), adversarial training with Gaussian-filter smoothing (G), adversarial training with embedded filter smoothing (E), and adversarial training with non-local Gaussian smoothing (NG). Results show that adversarially trained models (with smoothing filters) improve saliency map quality but at the expense of benign accuracy.

We illustrate this tradeoff in Figure 10 and Figure 11. We train $L_\infty(\epsilon)$ robust FMNIST and CIFAR-10 models with perturbation strength $\epsilon \in [0.01, 0.03, 0.06, 0.1]$ for adversarial training (A), adversarial training with smoothing filters of mean (M1), median (M2), Gaussian (G), embedded (E) and nonlocal Gaussian (NG). For each model, we compute its benign accuracy, and three saliency map characteristics using Vanilla Gradient: sparsity (Chalasani et al., 2020), area under perturbation curve of ROAD (Rong et al., 2022) for faithfulness, and structural similarity (Adebayo et al., 2018). Then, we plot the saliency map characteristics against the benign accuracy of the model.

As shown in Figure 10a, there is a clear inverse relationship between sparsity and benign accuracy in FMNIST models. Models trained with stronger adversarial perturbations (i.e., lower benign accuracy) tend

to produce sparser saliency maps. However, the trade-off in faithfulness is less straightforward, as seen in Figure 10b. While adversarial training with embedded and Gaussian filters (E, G) improves faithfulness for some robustness levels, there is no consistent trend across perturbation strengths. Interestingly, the naturally trained model (N) shows competitive or superior faithfulness at high accuracy. This reinforces the nuanced effect of adversarial training and smoothing on faithfulness, especially on datasets like FMNIST. Figure 10c illustrates that models trained with higher perturbation strengths produce saliency maps that are structurally more consistent under noise, but this again comes with a drop in benign accuracy.

On CIFAR-10, as shown in Figure 11a, we observe a strong inverse relationship between benign accuracy and saliency map sparsity, similar to FMNIST. The faithfulness plot in Figure 11b shows a distinct improvement in attribution faithfulness for adversarially trained models with smoothing filters. Notably, the embedded filter (E) and Gaussian filter (G) models peak in faithfulness at intermediate accuracy levels. The naturally trained model (N), despite its high accuracy, falls behind in faithfulness, suggesting that robust models—especially those with smoothing—better align attributions with model predictions. Figure 11c shows that structurally stable saliency maps (i.e., maps that are robust to noise perturbations) are more prominent in adversarially trained models. Most smoothing-based models achieve high SSIM, indicating stable saliency responses under input perturbations.

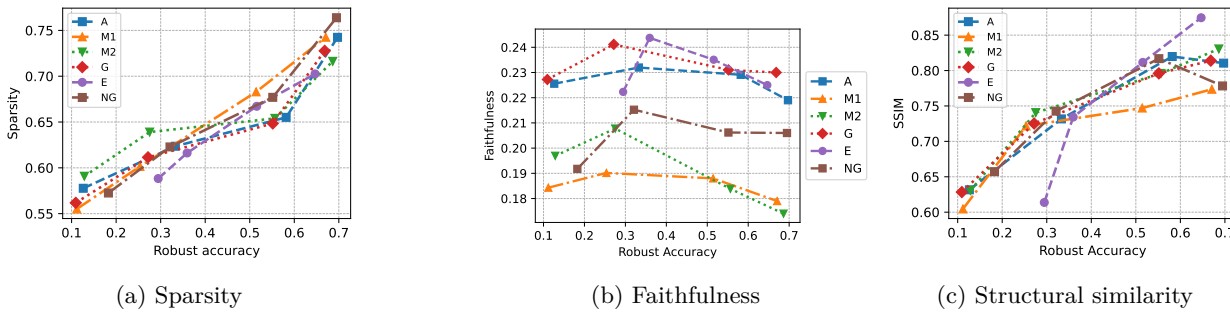

(a) Sparsity           (b) Faithfulness           (c) Structural similarity

Figure 12: Relationship between model robustness and saliency map quality on FMNIST models: naturally trained (N), adversarially trained (A), adversarial training with mean-filter smoothing (M1), adversarial training with median-filter smoothing (M2), adversarial training with Gaussian-filter smoothing (G), adversarial training with embedded filter smoothing (E), and adversarial training with non-local Gaussian smoothing (NG). Results show that increasing robustness of adversarially trained models (with smoothing filters) improves saliency map quality.

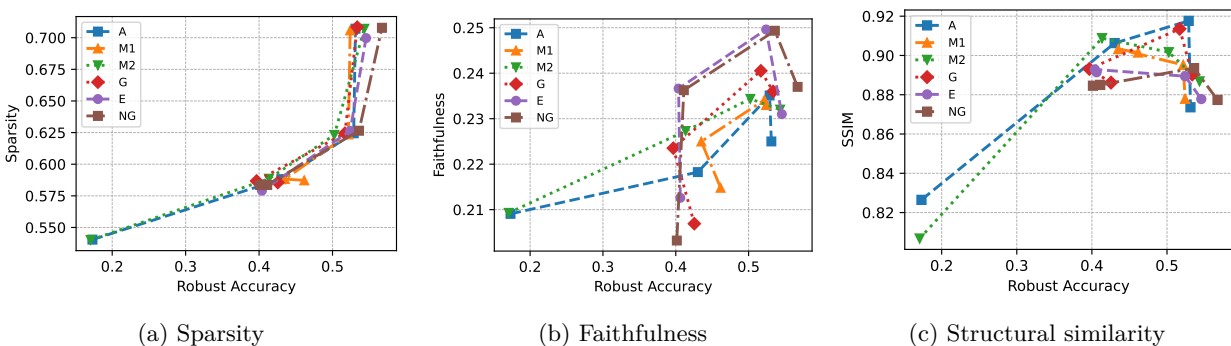

(a) Sparsity           (b) Faithfulness           (c) Structural similarity

Figure 13: Relationship between model robustness and saliency map quality on CIFAR-10 models: naturally trained (N), adversarially trained (A), adversarial training with mean-filter smoothing (M1), adversarial training with median-filter smoothing (M2), adversarial training with Gaussian-filter smoothing (G), adversarial training with embedded filter smoothing (E), and adversarial training with non-local Gaussian smoothing (NG). Results show that increasing robustness of adversarially trained models (with smoothing filters) improves saliency map quality.

### 4.2.5    Relationship between model robustness & saliency map quality

In Section 4.2.4, we observed that as the $\epsilon$ perturbation strength of adversarial training increases, the benign accuracy of the model decreases. However, this is accompanied by an improvement in the quality of saliency maps. In this section, we extend our analysis by examining the relationship between robust accuracy and the quality of saliency maps. As the $\epsilon$ perturbation strength increases, model robustness—defined as the ability to correctly classify adversarially perturbed inputs—also improves. For each $L_\infty(\epsilon)$ model trained at $\epsilon \in \{0.01, 0.03, 0.06, 0.1\}$, we compute its robust accuracy by evaluating its performance on PGD (Madry et al., 2018) adversarial examples, generated with $\epsilon = 0.1$ and 100 steps. Then, we plot the relationship between sparsity (Chalasani et al., 2020), faithfulness using ROAD area under perturbation curve (Rong et al., 2022), and structural similarity (Adebayo et al., 2018) against robust accuracy in Figure 12 and Figure 13.

As shown in Figure 12a, there is a strong positive correlation between robust accuracy and sparsity. Models with higher robustness—typically those trained with stronger perturbations—produce sparser saliency maps. The trend for faithfulness, for FMNIST, is more nuanced, as illustrated in Figure 12b. Some models, like those trained with embedded (E) and Gaussian (G) smoothing, exhibit peak faithfulness at moderate robustness levels but decline as robustness increases further. This suggests that while a certain level of robustness contributes to more faithful saliency maps, excessive robustness may hurt attribution alignment with model predictions. Figure 12c shows a clear positive correlation between robust accuracy and saliency map structural similarity. Models trained with stronger adversarial perturbations and smoothing filters yield saliency maps that are more structurally consistent under noise.

On CIFAR-10, we again observe a a clear and consistent positive relationship between robust accuracy and saliency map sparsity in Figure 13a. The faithfulness plot in Figure 13b shows that models with moderate-to-high robustness tend to yield more faithful explanations, especially when trained with smoothing filters. While some smoothing configurations (e.g., M1) show irregular fluctuations, the general trend suggests that faithfulness improves with robustness, especially when combined with appropriate smoothing. Figure 13c shows that the structural similarity of saliency maps under noise increasingly improves as model robust accuracy improves.

## 4.3    Qualitative Analysis

Our quantitative studies demonstrate that saliency maps in adversarially trained models are sparse but at the expense of stability. Incorporating local feature-map smoothing improves stability of saliency maps without drastically compromising sparsity, balancing these two aspects. In this section, we analyze how well end-users comprehend saliency maps from different model training strategies based on the level of sparsity.

**Motivation:** The goal of an explanation method is to provide insights into the model's reasoning process. While faithfulness is crucial, the comprehensibility of explanations to human users is equally important, particularly in decision-making contexts where AI models assist experts. Since saliency maps are used by human end-users, an explanation method must be both faithful and understandable to be effective. An explanation that accurately reflects model behavior but is too noisy or unclear may not be useful for practical decision-making (Gilpin et al., 2018). While prior works (Nguyen et al., 2021; Kim et al., 2022; Adebayo et al., 2020) have focused on qualitative evaluation for utility of explanations, we conduct a survey to measure comprehensibility of saliency maps.

**Survey Methodology:** We conducted an experiment with 65 graduate students (Ph.D./ Masters), each with at least a year of experience in computer vision[1]. The objective was to determine whether the information conveyed by saliency maps was sufficient for understanding and trusting the underlying model behavior. Participants were shown saliency maps using Vanilla Gradient from three models—naturally trained, adversarially trained, and adversarially trained with feature-map smoothing (median filter)—for 10 images across FMNIST and CIFAR-10 datasets, resulting in 60 image-saliency pairs. The saliency maps were presented in random order, and participants were unaware of the model that generated them. Afterward, they rated each

---

[1]An Institutional Review Board (IRB) approval was granted by our institution prior to interviewing human subjects for our qualitative study.

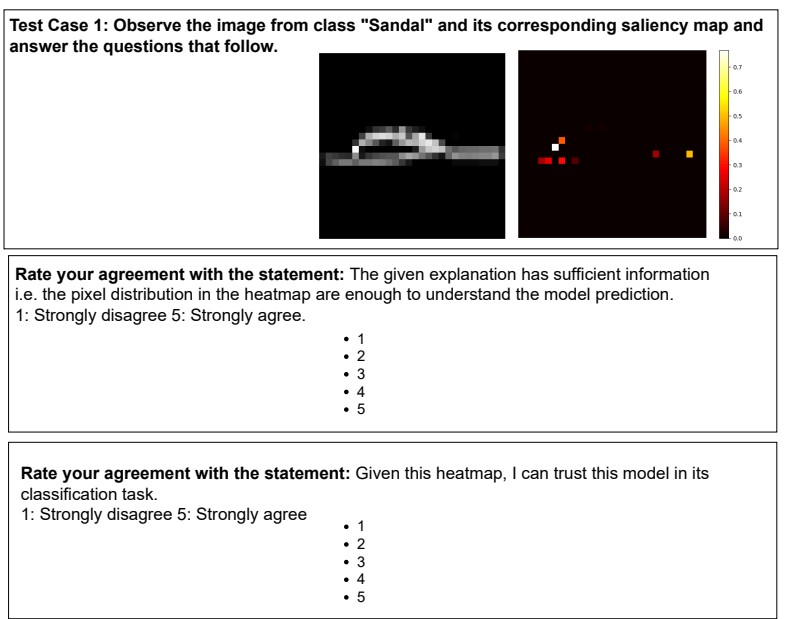

Figure 14: A sample of question from the survey.

saliency map using the Hoffman satisfaction scale (Hoffman et al., 2023), responding to two key questions: 1) "Does the explanation provide sufficient information?" and 2) "Do you trust the model's classification based on this saliency map?" Ratings were on a scale of 1 (strongly disagree) to 5 (strongly agree) (See Figure 14 for a sample). Finally, participants were asked to compare saliency maps from all three models side by side (as shown in Figure 15) and select the most comprehensible explanation, providing free-text justifications for their choices.

**Results:** We assessed the comprehensibility of the saliency maps based on two metrics: sufficiency and trust. Sufficiency metric corresponds to the survey question "Does the given explanation have sufficient information?" and trust corresponds to the survey question "Given this heatmap, do you trust the model's classification?". For the naturally trained model, participants rated sufficiency at an average of 2.08 ($\pm$ 0.75) and trust at 2.02 ($\pm$ 0.82), indicating that the noisy maps from this model were generally considered untrustworthy. In contrast, adversarially trained models fared better, with sufficiency scoring 2.99 ($\pm$ 0.93) and trust 3.08 ($\pm$ 0.90), as participants found these maps clearer and more aligned with the images. The feature-map smoothed adversarial model scored the highest, with sufficiency at 3.33 ($\pm$ 1.03) and trust at 3.14 ($\pm$ 1.01). Participants appreciated the reduction in noise and highlighted the clarity and relevance of the explanations. When comparing saliency maps directly, 56% of participants preferred the maps from the feature-map smoothed model, 29% favored the adversarial model, and only 15% selected the naturally trained model. The majority cited reasons such as "highlighting important features without excessive detail" and "close enough to the image with the least noise".

To ensure that the observed differences in participant ratings were statistically meaningful rather than random variations, we performed Wilcoxon signed-rank tests Woolson (2007) and one-way ANOVA Cuevas et al. (2004). The Wilcoxon test assesses whether paired differences between two conditions (e.g., adversarially trained vs. naturally trained) are statistically significant, making it well-suited for analyzing subjective survey responses. The one-way ANOVA test determines whether there are significant differences across all three models. As shown in Table 2, the extremely small p-values ($< 0.001$) indicate that differences in both sufficiency and trust scores across training strategies are statistically significant. This supports our claim that adversarial training and feature-map smoothing significantly improve the interpretability of saliency maps.

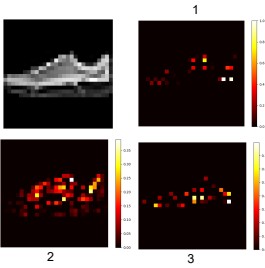

Figure 15: Sample image and saliency maps used in the survey.

| | Wilcoxon (p-value) | | | one-way ANOVA | |
|---|---|---|---|---|---|
| | N vs A | A vs M2 | N vs M2 | F-stat | p value |
| **Sufficiency** | 9.79E-41 | 4.26E-14 | 3.71E-27 | 200.38 | 7.82E-72 |
| **Trust** | 5.56E-39 | 3.24E-11 | 3.89E-24 | 193.86 | 6.58E-70 |

Table 2: Wilcoxon and ANOVA test results on the survey where, N refers to a naturally trained model, A refers to an adversarially-trained model, and M2 referes an adversarial-trained feature-map smoothed model.

## 5 Limitations

This work focuses exclusively on input-gradient-based explanation methods, evaluating the sparsity-stability trade-off using three widely adopted techniques: Vanilla Gradient (VG) (Simonyan et al., 2014), Integrated Gradients (IG) (Sundararajan et al., 2017), and SmoothGrad (SG) (Smilkov et al., 2017). While these methods are representative of the broader class of gradient-based explainability techniques, they do not encompass the full spectrum of interpretability approaches. In future work, we plan to extend our analysis to include other backpropagation-based methods and perturbation-based approaches.

We conduct our experiments on three widely used image classification datasets—FMNIST, CIFAR-10, and ImageNette. However, scaling adversarial training to more complex models and larger datasets, particularly those with higher class counts, remains a significant challenge (Zhang et al., 2019). Achieving a balance between high accuracy and robustness in such settings presents a significant challenge and may affect the generalizability of our findings.

Moreover, while we explore a range of local and non-local feature map smoothing techniques, the selection of the optimal smoothing strategy and its hyperparameters remains empirical and task-dependent. A more principled approach to selecting or learning such filters is a potential direction for future work.

Lastly, our theoretical insights are grounded in a simplified modeling of neural networks as single-layer systems. While this abstraction aids analytical clarity, aligning with prior works (Chalasani et al., 2020), it does not fully capture complex inter-layer interactions and non-linearities present in the deep architectures, and its direct applicability to highly non-linear deep networks should be interpreted with caution.

## 6 Conclusion

In this paper, we explore the connection between model training strategies and quality of explanations, and propose a simple modification to adversarial training to improve the comprehensibility of saliency maps. Through a comprehensive study, we established that the quality of saliency maps is tied to the sensitivity of a model, with adversarially trained models producing sparser but unstable explanations. Incorporating local feature-map smoothing during adversarial training enhances stability and comprehensibility without sacrificing sparsity. By shedding light on the trade-offs between robustness of a model and saliency map quality, we advocate for the designing models that strike a balance between performance and saliency map comprehensibility.

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

## A    Dataset and Training

**FMNIST (Xiao et al., 2017):** The Fashion MNIST dataset consists of 28x28 pixel grayscale images of various clothing items and accessories. It contains a total of 70,000 images, divided into a training set of 60,000 examples and a test set of 10,000 examples. Similar to Chalasani et al. (2020), we train a neural network consisting of two convolutional layers with 32 and 64 filters, respectively, each followed by 2x2 max-pooling and a fully connected layer of 1024. We use the Adam optimizer with a learning rate of 0.001, a batch size of 32 and 50 training epochs.

**CIFAR-10 (Krizhevsky et al., 2009):** CIFAR-10 consists of 60,000 32x32 pixel color images, with each image belonging to one of ten different classes. These classes include common objects and animals such as airplanes, automobiles, birds, cats, deer, dogs, frogs, horses, ships, and trucks. Similar to Chalasani et al. (2020), we use a wide Residual Network (Zagoruyko & Komodakis, 2016) for training CIFAR-10 with the following hyperparameter settings: batch size=128, momentum optimizer with momentum = 0.9, and weight decay = 5e-4, training steps = 70000. We use an adaptive learning rate where the learning rate is set to 0.1 for the first 40000 steps, 0.01 for 40000-50000 steps, and 0.001 for the remaining steps. The wide residual network is trained with 28 layers and widen factor of 10.

**ImageNette (Howard, 2020):** ImageNette is a 10-class subset of ImageNet (Deng et al., 2009) with 9469 training images and 3925 test images. We use the 320-pixel resolution images (for the shortest side) and randomly resize and crop them to 224x224 pixels during training. We use the standard ResNet-18 model architecture for training on the dataset. We use Ranger optimizer (Wright, 2019) with an initial learning rate of 8e-03 and epsilon 1e-6. We train the models from scratch for 200 epochs and employ the early stopping criterion to select the best-performing model for evaluation.

### A.1    Adversarial training

Adversarial training (Goodfellow et al., 2015) is a machine learning technique that involves training a model in the presence of adversarial examples. Adversarial examples are inputs specifically designed to mislead or deceive the model, causing the model to make incorrect predictions. The goal of adversarial training is to improve the robustness and generalization of a model against such perturbed examples. To perform adversarial training, we generate adversarial examples that are produced from natural samples $\mathbf{x} \in R^d$ by adding a perturbation vector $\delta \in R^d$. The perturbation vector differs based on the type of attack employed. We use the PGD (Madry et al., 2018) attack to obtain adversarial perturbations. PGD is an iterative attack where the perturbation is computed multiple times with small steps. The hyper-parameters of PGD attack in our adversarial training: for FMNIST and CIFAR-10, $\epsilon \in \{0.01, 0.03, 0.06, 0.1\}$, attack step size $= \epsilon/10$, and number of iterations = 40; for ImageNette $\epsilon = 1/255$, step size $= 0.00784$ and number of iterations = 20. Other training hyperparameters are kept as explained in Appendix A.

## B    Smoothing filters

A generic convolutional neural network with a feature map smoothing block is presented in Figure 16. The smoothing block consists of local or non-local filtering operations. All feature-map smoothed models are trained with the same hyper-parameter settings as explained in Appendix A. We use with the following filters in the paper:

### B.1    Local smoothing:

Local smoothing applies filtering operations to a neighborhood of a feature map. We use the following local smoothing filters in our approach:

- **Mean filter:** A mean filter, equivalent to an average pooling with a stride of 1, replaces each feature with the average of nearby features within a defined kernel. This smoothing effect reduces noise and enhances robustness to spatial variations. For an input feature map ($I$) of size $H$x$W$ and a $K$-sized

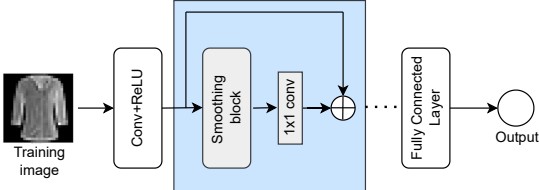

Figure 16: A generic convolutional neural network with a feature-map smoothing block.

kernel, the output feature map $O(u,v)$ is calculated using Eqn. 9:

$$O(u,v) = \frac{1}{K^2} \sum_{i=0}^{K-1} \sum_{j=0}^{K-1} I(u+i, v+j) \tag{9}$$

Here, $u$ and $v$ represent spatial coordinates in the output feature map, ranging from 0 to $H - K$ and 0 to $W - K$ respectively. $I(u+i, v+j)$ denotes the feature value at spatial location $(u+i, v+j)$ in the input feature map. This operation is applied independently to each channel of the input feature map.

- **Median filter:** A median filter, unlike a mean filter, computes the median value within a small sliding window over the feature map, given by Eqn. 10. This method also removes noise, making representations more robust. It preserves edges and fine details as it selects the median value. Given an input feature map $I$ and a median filter window size $K$, the output feature map $O(u,v)$ is computed using Eqn. 10:

$$O(u,v) = median(I(u - \frac{K}{2} : u + \frac{K}{2}, v - \frac{K}{2} : v + \frac{K}{2}) \tag{10}$$

Here, $I(u - \frac{K}{2} : u + \frac{K}{2}, v - \frac{K}{2} : v + \frac{K}{2})$ represents the subset of the input feature around $(u,v)$ with a size of $KxK$. This operation is applied independently to each channel of the input feature map. Since median filters are non-linear and non-differentiable operations, this can pose challenges when training a neural network end-to-end. We utilize the approximation of the median filter available in Kornia (Riba et al., 2020), which is differentiable.

- **Gaussian filter:** A Gaussian filter applies a smoothing effect to feature maps by convolving them with a Gaussian kernel, effectively reducing Gaussian noise. This process improves the signal-to-noise ratio and preserves edges better than mean filtering due to the Gaussian kernel giving more weight to nearby features while still considering distant feature contributions. The degree of smoothing can be adjusted by modifying the standard deviation ($\sigma$) of the Gaussian kernel. Given an input feature map $I$ and a Gaussian filter kernel $K$, the output feature map $O(u,v)$ is calculated with Eqn. 11:

$$O(u,v) = (I * K)(u,v) \tag{11}$$

Here, $*$ denotes 2D convolution. The Gaussian kernel $K$ is generated using a Gaussian function with a specific standard deviation $\sigma$, defined in Eqn. 12:

$$K(u,v) = \frac{1}{2\pi\sigma^2} e^{(-\frac{u^2+v^2}{2\sigma^2})} \tag{12}$$

This operation is independently applied to each channel of the input feature map.

**Implementation:** We utilize the differentiable filters available in Kornia (Riba et al., 2020). We use a 3x3 Kernel for mean, median, and Gaussian filtering. The standard deviation of the kernel for Gaussian filtering was computed as $(0.3 * ((\mathbf{x}.shape[3] - 1) * 0.5 - 1) + 0.8, 0.3 * ((\mathbf{x}.shape[2] - 1) * 0.5 - 1) + 0.8)$ where $\mathbf{x}$ is the input image.

## B.2 Non-local smoothing:

The non-local approach (Buades et al., 2005) derives a smooth feature map $m$ from an input feature map $x$ by calculating a weighted average of features across all spatial positions within the set $\mathcal{L}$. Eqn. 13 shows the formulation where $f(x_i, x_j)$ is feature dependent weighting function and $\mathcal{C}(x)$ is a normalization function.

$$m_i = \frac{1}{\mathcal{C}(x)} \sum_{\forall j \in \mathcal{L}} f(x_i, x_j).x_j \tag{13}$$

We consider the following forms of weighting function $f(.)$:

- **Non-local Gaussian (Wang et al., 2018):** Eqn. 14 formulates the non-local gaussian function where $x_i^T x_j$ is the dot product similarity between the feature maps. The normalization function is set as $\mathcal{C}(x) = \sum_{\forall x} f(x_i, x_j)$.

$$f(x_i, x_j) = e^{(x_i^T x_j)} \tag{14}$$

- **Embedded Gaussian (Wang et al., 2018):** This non-local mean computes similarity in embedding space by computing embedded versions of the feature map $x$. As shown in Eqn. 15, $\theta(x_i) = W_\theta x_i$ and $\eta(x_j) = W_\phi x_j$ are the two embeddings of feature map $x$, obtained after $1 \times 1$ convolution. The normalization function is set as $\mathcal{C}(x) = \sum_{\forall x} f(x_i, x_j)$.

$$f(x_i, x_j) = e^{(\theta(x_i)^T \eta(x_j))} \tag{15}$$

We use the open-source implementation of non-local means available in AlexHex7 (2018).

Table 3: Natural and Robust Accuracy of Various FMNIST, CIFAR-10, and ImageNette models: naturally trained (N), adversarially trained (A), natural training with mean-filter smoothing (M1), adversarial training with mean-filter smoothing (M1+A), natural training with median-filter smoothing (M2), adversarial training with median-filter smoothing (M2+A), natural training with Gaussian-filter smoothing (G), adversarial training with Gaussian-filter smoothing (G+A), natural training with embedded filter smoothing (E), adversarial training with embedded filter smoothing (E+A), natural training with non-local Gaussian smoothing (NG), and adversarial training with non-local Gaussian smoothing (NG+A).

| Dataset | Models/Accuracy | N | A | M1 | M1+A | M2 | M2+A | G | G+A | E | E+A | NG | NG+A |
|---|---|---|---|---|---|---|---|---|---|---|---|---|---|
| **FMNIST** | **Benign Accuracy** | 89.9 | 79.9 | 88.4 | 80.0 | 88.8 | 80.5 | 89.1 | 80.3 | 89.4 | 81.1 | 89.23 | 81.3 |
| | **Robust Accuracy** | 9.5 | 67.7 | 8.5 | 67.1 | 8.2 | 68.6 | 6.9 | 66.8 | 7.31 | 64.7 | 7.23 | 69.5 |
| **CIFAR-10** | **Benign Accuracy** | 90.9 | 80.5 | 89.7 | 79.6 | 88.6 | 80.1 | 90.2 | 80.8 | 90.6 | 79.6 | 89.9 | 81.9 |
| | **Robust Accuracy** | 4.8 | 54.3 | 4.5 | 51.2 | 4.7 | 56.3 | 6.8 | 53.9 | 5.1 | 55.5 | 7.1 | 55.8 |
| **ImageNette** | **Benign Accuracy** | 96.3 | 70.8 | 93.3 | 58.8 | 90.9 | 55.3 | 95.5 | 51.6 | 88.4 | 60.8 | 86.3 | 58.4 |
| | **Robust Accuracy** | 1.6 | 12.2 | 1.2 | 6.5 | 2.3 | 14.3 | 3.7 | 13.5 | 3.1 | 13.9 | 2.5 | 18.9 |

# C   Effect of smoothing filter

In Table 3, we present the results of various models on FMNIST, CIFAR-10 and ImageNette, with both natural (benign) and adversarial (robust) accuracy. Benign accuracy measures the model performance on benign (clean) test set, whereas robust accuracy evaluates how well the models detect adversarially perturbed samples. The robust models under evaluation are trained at $\epsilon = 0.1$ for FMNIST and CIFAR-10 and $\epsilon = 1/255$ for ImageNette. Evaluation is performed on a test-set consisting of adversarial samples created using PGD attack (Madry et al., 2018) at $\epsilon = 0.1$ $l_\infty$ perturbation bound.

Across all datasets, applying smoothing filters alone did not result in significant changes in natural or robust accuracy ($\approx \pm 3\%$). The smoothing filters, when used without adversarial training, did not drastically

Table 4: Sparsity and Stability evaluation, when smoothing block is placed after second residual block, on various CIFAR-10 models: adversarial training with mean-filter smoothing (M1), adversarial training with median-filter smoothing (M2), adversarial training with Gaussian-filter smoothing (G), adversarial training with embedded filter smoothing (E), and adversarial training with non-local Gaussian smoothing (NG)

|  | CIFAR-10 | | | | |
|---|---|---|---|---|---|
|  | M1 | M2 | G | E | NG |
| dG (higher the better) | 0.178 | 0.185 | 0.176 | 0.190 | 0.191 |
| dRIS (lower the better) | -0.605 | -0.663 | -0.477 | -0.528 | -0.621 |
| dROS (lower the better) | 0.268 | 0.225 | 0.239 | 0.273 | 0.269 |
| dRRS (lower the better) | 0.464 | 0.445 | 0.462 | 0.453 | 0.475 |

Table 5: Sparsity and Stability evaluation, when smoothing block is placed after third residual block, on various CIFAR-10 models: adversarial training with mean-filter smoothing (M1), adversarial training with median-filter smoothing (M2), adversarial training with Gaussian-filter smoothing (G), adversarial training with embedded filter smoothing (E), and adversarial training with non-local Gaussian smoothing (NG)

|  | CIFAR-10 | | | | |
|---|---|---|---|---|---|
|  | M1 | M2 | G | E | NG |
| dG (higher the better) | 0.185 | 0.180 | 0.187 | 0.191 | 0.192 |
| dRIS (lower the better) | -0.599 | -0.670 | -0.470 | -0.517 | -0.612 |
| dROS (lower the better) | 0.271 | 0.221 | 0.235 | 0.276 | 0.261 |
| dRRS (lower the better) | 0.470 | 0.429 | 0.468 | 0.446 | 0.473 |

improve robustness or reduce natural accuracy, indicating that their primary role may be in stabilizing feature maps without dramatically altering decision boundaries.

However, when smoothing filters were combined with adversarial training, robust accuracy improved for some filters, particularly in FMNIST and CIFAR-10, where models trained with adversarial samples and smoothing exhibited stronger defense against adversarial attacks. On the ImageNette dataset, we observed a notable drop in benign accuracy when smoothing filters were applied during adversarial training.

## D  Ablation study: Position of smoothing filters

In this section, we investigate how the placement of smoothing filters within the network affects the stability and sparsity of saliency maps. Specifically, we consider different positions for inserting the smoothing filters in a CIFAR-10 network and report the results in Tables 4 and 5 for Vanilla Gradient (Simonyan et al., 2014). This CIFAR-10 Residual Network consists of three residual blocks. We add smoothing filters after second residual block in Table 4 and after third residual block in Table 5. In Table 1, smoothing filters are added after the first residual block.

Across all Tables, the sparsity gain remains consistent between 0.176 to 0.192; however, when smoothing filter is added after third residual block (Table 5), there is a slight improvement in the sparsity. Smoothing after the first block consistently yields better results in stability. Hence, to strike a balance between stability and sparsity, we place the smoothing block after the first residual block.

## E  Additional experiment

In this section, we demonstrate the effects of robust training strategy on saliency map quality for a different network, VGG16 (Simonyan & Zisserman, 2015) on CIFAR-10. We train a VGG-16 convolutional neural network for 120 epochs using stochastic gradient descent (SGD) with momentum, a learning rate of 0.1, and weight decay of 5e-4. The model consists of five convolutional blocks with batch normalization, ReLU activations, max-pooling layers, and a fully connected classifier. The training utilizes a learning rate scheduler, which reduces the learning rate by a factor of 0.1 every 30 epochs. For adversarial training, we use the same hyperparameter (PGD attack at $\epsilon = 0.1$). The hyperparameters for smoothing blocks are also kept as discussed in Appendix B. Similar to previous sections, we train following models for this network: naturally-trained (N), adversarially-trained (A), adversarial training with mean-filter smoothing (M1), adversarial training with median-filter smoothing (M2), adversarial training with Gaussian-filter smoothing (G), adversarial training with embedded filter smoothing (E), and adversarial training with non-local gaussian smoothing (NG).

Table 6: Sparsity and Stability Evaluations for VG, IG, and SG on various VGG-16 models: adversarially-trained (A), adversarial training with mean-filter smoothing (M1), adversarial training with median-filter smoothing (M2), adversarial training with Gaussian-filter smoothing (G), adversarial training with embedded filter smoothing (E), and adversarial training with non-local Gaussian smoothing (NG). Here, ↑ and ↓ indicate higher and lower values are better.

| | Vanilla Gradient (VG) | | | | | | Integrated Gradients (IG) | | | | | | SmoothGrad (SG) | | | | | |
|---|---|---|---|---|---|---|---|---|---|---|---|---|---|---|---|---|---|---|
| | A | M1 | M2 | G | E | NG | A | M1 | M2 | G | E | NG | A | M1 | M2 | G | E | NG |
| **dG** ↑ | 0.10 | 0.10 | 0.10 | 0.10 | 0.11 | 0.09 | 0.02 | 0.03 | 0.02 | 0.01 | 0.01 | 0.02 | 0.08 | 0.08 | 0.08 | 0.08 | 0.08 | 0.08 |
| **dRIS** ↓ | -0.30 | -0.40 | -0.35 | -0.39 | -0.39 | -0.42 | -0.29 | -0.62 | -0.74 | -0.60 | -0.84 | -0.81 | -0.33 | -0.36 | -0.46 | -0.10 | -0.49 | -0.52 |
| **dROS** ↓ | -0.24 | -0.31 | -0.26 | -0.30 | -0.30 | -0.32 | -0.13 | -0.22 | -0.52 | -0.24 | -0.52 | -0.56 | -0.42 | -0.50 | -0.49 | -0.40 | -0.47 | -0.53 |
| **dRRS** ↓ | 0.28 | 0.21 | 0.25 | 0.19 | 0.19 | 0.18 | 0.24 | 0.17 | -0.25 | 0.04 | -0.35 | -0.24 | 0.06 | 0.03 | 0.02 | 0.05 | 0.01 | -0.09 |

Next to evaluate sparsity, and stability, for each model, we compute explanations using Vanilla Gradient (VG), Integrated Gradients (IG), and SmoothGrad (SG), and then compute its sparseness using Gini index (G) (Chalasani et al., 2020), and its stability using relative input stability (RIS), relative output stability (ROS) and relative representation stability (RRS) (Agarwal et al., 2022). Similar to Chalasini et al. (Chalasani et al., 2020), we compare the sparsity and stability improvement of saliency maps with respect to the naturally trained model (N). Specifically, for a given training method (M), we compute the following metrics that quantify the improvement in sparseness (dG), relative input stability (dRIS), relative output stability (dROS), and relative representation stability (dRRS) of the explanation method $\phi(.) \in \{VG, IG, SG\}$:

$$dG[\phi(\mathbf{x})] = G^M[\phi(\mathbf{x})] - G^N[\phi(\mathbf{x})] \tag{16}$$

$$dRIS[\phi(\mathbf{x})] = RIS^M[\phi(\mathbf{x})] - RIS^N[\phi(\mathbf{x})] \tag{17}$$

$$dROS[\phi(\mathbf{x})] = ROS^M[\phi(\mathbf{x})] - ROS^N[\phi(\mathbf{x})] \tag{18}$$

$$dRRS[\phi(\mathbf{x})] = RRS^M[\phi(\mathbf{x})] - RRS^N[\phi(\mathbf{x})] \tag{19}$$

Table 6 shows the results of sparsity and stability evaluation of saliency maps generated by Vanilla Gradient (VG), Integrated Gradients (IG), and SmoothGrad (SG) across a variety of models in VGG network. We can observe that all explanation methods show positive dG values across all models, indicating that the saliency maps become sparser when used with robust, adversarially trained VGG models. The sparsity gain, however, remains relatively stable across models, with only slight variations. This suggests that while robust training introduces sparsity, the choice of smoothing filter does not significantly impact the sparsity of explanations.

In terms of input and output stability ($dRIS$ and $dROS$), we observe that models enhanced with smoothing filters (M1, M2, G, E, NG) consistently exhibit better stability compared to the adversarially trained baseline (A). This is particularly pronounced in the IG and SG methods, where stability improvements are more significant. The introduction of smoothing filters, such as median and Gaussian, mitigates the instability of explanations seen in the baseline model, resulting in more stable and reliable saliency maps.

## F   Conditions Affecting the Tightness of Stability Bounds

The stability bounds presented in Section 3.1 serve as indicators of the relationship between model sensitivity and attribution stability. However, these bounds are inherently approximate and depend on several factors. For example, the nonlinearity of the model, particularly the choice of activation function $H$, might influence the bounds' tightness. For activation functions with bounded gradients, such as sigmoid or tanh, the change in $H'(\langle \mathbf{w}, \mathbf{x} \rangle)$ is limited, leading to more consistent attributions across small perturbations and therefore tighter stability bounds. Specifically, for sigmoid, $H(z) = \frac{1}{1+e^{-z}}$ and $H'(z) = H(z)(1 - H(z))$, both of which remain bounded as $H(z)$ approaches 0 or 1. Conversely, for ReLU activation, $H(z) = \max(0, z)$ with $H'(z) = 1$ when $z > 0$ and 0 otherwise, the gradient can change abruptly across input perturbations. Thus, for perturbations where $\mathbf{x}$ is shifted across the activation boundary, $H'(\langle \mathbf{w}, \mathbf{x} \rangle)$ may vary significantly, producing looser bounds. Similarly, the type and scale of input perturbations (Gaussian noise with $\mathbf{n} \sim \mathcal{N}(0, \sigma^2)$) can also impact

bound tightness. For small perturbations with minimal output change, the stability bounds remain tight. However, larger perturbations can result in more significant output shifts $|F(\mathbf{x}') - F(\mathbf{x})|$, leading to looser bounds. This can be pronounced for high dimensional images which tend to lie close to decision boundaries, making them susceptible to small noise that can lead to misclassification (Tanay & Griffin, 2016). In addition, weight regularization techniques, such as weight decay, result in smoother gradients, reducing the sensitivity of $F(\mathbf{x})$ to input changes. For instance, regularized models with smaller gradient norms can have tighter stability bounds as $H'(\langle \mathbf{w}, \mathbf{x} \rangle) \cdot \mathbf{w}$ varies less across the input space. Lastly, datasets with high intraclass variability introduce more variable responses to perturbations, increasing $|F(\mathbf{x}') - F(\mathbf{x})|$. As a result, stability bounds may become looser due to the variability in $F(\mathbf{x})$ across samples.

## G    Study on receptive field expansion

To measure the receptive field effect in the smoothing block, we conduct an additional experiment on CIFAR-10 where we modify the feature smoothing block so that it performs only a convolution (identify or randomly initialized). This modified setup ensures that there is only an expansion of the receptive field without filtering operations and it can provide a baseline study to analyze the effect of receptive field expansion on its own. Table 7 shows the results for Vanilla Gradient (VG) when compared with the best performing model.

Table 7: Sparsity and Stability evaluation for Vanilla Gradients. *Here, M2: adversarial training with median smoothing, Identity: adversarial training with feature smoothing block consisting of identify convolution but no smoothing filter and Random: adversarial training with feature smoothing block consisting of randomly initialized convolution but no smoothing filters*

| Models | M2 | Identity | Random |
|---|---|---|---|
| Sparsity (dG) (hgiher is better) | 0.18 | 0.16 | 0.15 |
| Relative input stability (dRIS) (lower is better) | -0.68 | -0.41 | -0.36 |
| Relative output stability (dROS) (lower is better) | 0.21 | 0.07 | 0.06 |
| Relative representation stability (dRRS) (lower is better) | 0.43 | 0.41 | 0.43 |

The results in the table show tha the 'M2' model still achieves the best sparsity, indicating that the smoothing operation in addition to the convolutional operation helps the model to learn a smaller number of discriminative features. The 'M2' model also performs significantly better in input stability. This indicates that smoothing filters provide stability in saliency maps with respect to input. Interestingly, the 'M2' model does not achieve the best score in output stability. This suggests that while smoothing helps in stabilizing attributions with respect to inputs and internal representations, it might not directly translate to stability at the model's output layer. The expanded receptive field introduced by the identity or random convolutions likely contributes to this improvement. The 'Identity' model achieves the best representation stability but only marginally outperforming 'M2'. Overall, the inclusion of smoothing operations still provides a competitive advantage in improving the quality of saliency maps with respect to sparsity, input stability and representation stability.

## H    Relationship between attribution stability and model sensitivity

Consider a single-layer DNN with the form $F(\mathbf{x}) = H(\langle \mathbf{w}, \mathbf{x} \rangle)$, where $H$ is a differentiable scalar-valued activation function (e.g., sigmoid), $\langle \mathbf{w}, \mathbf{x} \rangle$ is the dot product between the weight vector $\mathbf{w}$ and input $\mathbf{x} \in \mathbb{R}^d$.

### H.1    Relationship for Vanilla Gradient (VG)(Simonyan et al., 2014)

Let $\mathbf{x} \in R^d$ denote an input image. The Vanilla Gradient (VG) explanation for a model $F$ is computed as,

$$VG(\mathbf{x}) = \frac{\partial F_c(\mathbf{x})}{\partial \mathbf{x}} \tag{20}$$

For a single-layer DNN with the form $F(\mathbf{x}) = H(\langle \mathbf{w}, \mathbf{x} \rangle)$, where $H$ is a differentiable scalar-valued activation function, $\langle \mathbf{w}, \mathbf{x} \rangle$ is the dot product between the weight vector $\mathbf{w}$ and input $\mathbf{x} \in \mathbb{R}^d$, the VG can be computed by applying the chain rule as follows:

$$VG(\mathbf{x}) = \frac{\partial H(\langle \mathbf{w}, \mathbf{x} \rangle)}{\partial \langle \mathbf{w}, \mathbf{x} \rangle} . \frac{\partial \langle \mathbf{w}, \mathbf{x} \rangle}{\partial \mathbf{x}} = H'(\langle \mathbf{w}, \mathbf{x} \rangle)\mathbf{w} \tag{21}$$

Here, $H'(\langle \mathbf{w}, \mathbf{x} \rangle)$ is the gradient of activation function $H$ with respect to the $\langle \mathbf{w}, \mathbf{x} \rangle$. Let $z = \langle \mathbf{w}, \mathbf{x} \rangle$ and $H(z) = \frac{1}{1+exp(-z)}$ be a sigmoid activation function then,

$$\begin{aligned} H'(z) &= \frac{exp(-z)}{(1+exp(-z))^2} \\ &= \frac{1}{1+exp(-z)}(1 - \frac{1}{1+exp(-z)}) \\ &= H(z)(1 - H(z)) \end{aligned} \tag{22}$$

Then, the VG attribution for an input $\mathbf{x}$ is given by

$$VG^F(\mathbf{x}) = H(\langle \mathbf{w}, \mathbf{x} \rangle)(1 - H(\langle \mathbf{w}, \mathbf{x} \rangle))\mathbf{w} \tag{23}$$

Now consider $\mathbf{x}' \in \mathcal{N}_\mathbf{x}$ is a noisy version of input image $\mathbf{x}$ where $\mathcal{N}_\mathbf{x}$ indicates a neighborhood of inputs $\mathbf{x}$ where the model prediction is locally consistent. Then, the VG attribution for an input $\mathbf{x}'$ is given by

$$VG^F(\mathbf{x}') = H(\langle \mathbf{w}, \mathbf{x}' \rangle)(1 - H(\langle \mathbf{w}, \mathbf{x}' \rangle))\mathbf{w} \tag{24}$$

The stability of the VG attribution is computed as the norm of the difference between the attribution of the original image and its noisy counterpart and can be expressed as

$$\Delta = ||VG^F(\mathbf{x}') - VG^F(\mathbf{x})||_1 \tag{25}$$

Substituting the expressions for $VG^F(\mathbf{x})$ and $VG^F(\mathbf{x}')$, and simplifying, we obtain

$$\begin{aligned} \Delta &= ||VG^F(\mathbf{x}') - VG^F(\mathbf{x})||_1 \\ &= ||H(\langle \mathbf{w}, \mathbf{x}' \rangle)(1 - H(\langle \mathbf{w}, \mathbf{x}' \rangle))\mathbf{w} - H(<\mathbf{w}, \mathbf{x}>)(1 - H(\langle \mathbf{w}, \mathbf{x} \rangle)).\mathbf{w}||_1 \\ &= ||\Big(H(\langle \mathbf{w}, \mathbf{x}' \rangle)(1 - H(\langle \mathbf{w}, \mathbf{x}' \rangle)) - H(\langle \mathbf{w}, \mathbf{x} \rangle)(1 - H(\langle \mathbf{w}, \mathbf{x} \rangle))\Big)\mathbf{w}||_1 \\ &= ||\Big(F(\mathbf{x}')(1 - F(\mathbf{x}')) - F(\mathbf{x})(1 - F(\mathbf{x}))\Big)\mathbf{w}||_1 \\ &= ||\Big((F(\mathbf{x}') - F(\mathbf{x}))(1 - F(\mathbf{x}') - F(\mathbf{x}))\Big)\mathbf{w}||_1 \end{aligned} \tag{26}$$

Bounding this by the magnitude of the change in model prediction,

$$\begin{aligned} \Delta &\le ||\Big(F(\mathbf{x}') - F(\mathbf{x})\Big)\mathbf{w}||_1 \\ \Delta &\le ||F(\mathbf{x}') - F(\mathbf{x})||_1.||\mathbf{w}||_1 \end{aligned} \tag{27}$$

Assuming $\mathbf{w}$ to be constant for a given model, the stability of the VG attribution is a direct result of the sensitivity of the model $||F(\mathbf{x}') - F(\mathbf{x})||$.

### H.2 Relationship for Integrated Gradients (IG) (Sundararajan et al., 2017)

The feature attribution score computed by Integrated Gradients (IG) for feature $i$ of input image $\mathbf{x} \in R^d$ with baseline $\mathbf{u}$, model $F$ is given by:

$$IG_i^F(\mathbf{x}, \mathbf{u}) = (x_i - u_i) . \int_{\alpha=0}^{1} \partial_i F(\mathbf{u} + \alpha(\mathbf{x} - \mathbf{u})) \partial \alpha \tag{28}$$

For an input image $\mathbf{x}$, IG returns a vector $IG^F(\mathbf{x}, \mathbf{u}) \in R^d$ with scores that quantify the contribution of $x_i$ to the model prediction $F(\mathbf{x})$. For a single layer network $F(\mathbf{x}) = H(\langle \mathbf{w}, \mathbf{x} \rangle)$ where $H$ is a differentiable scalar-valued function and $\langle \mathbf{w}, \mathbf{x} \rangle$ is the dot product between the weight vector $\mathbf{w}$ and input $\mathbf{x} \in R^d$, IG attribution has a closed form expression (Chalasani et al., 2020).

For given $\mathbf{x}$, $\mathbf{u}$ and $\alpha$, let us consider $\mathbf{v} = \mathbf{u} + \alpha(\mathbf{x} - \mathbf{u})$. If the single-layer network is represented as $F(\mathbf{x}) = H(\langle \mathbf{w}, \mathbf{x} \rangle)$ where $H$ is a differentiable scalar-valued function, $\partial_i F(\mathbf{v})$ can be computed as:

$$
\begin{aligned}
\partial_i F(\mathbf{v}) &= \frac{\partial F(\mathbf{v})}{v_i} \\
&= \frac{\partial H(\langle \mathbf{w}, \mathbf{v} \rangle)}{\partial v_i} \\
&= H'(z) \frac{\partial \langle \mathbf{w}, \mathbf{v} \rangle}{\partial v_i} \\
&= w_i H'(z)
\end{aligned}
\tag{29}
$$

Here, $H'(z)$ is the gradient of the activation $H(z)$ where $z = \langle \mathbf{w}, \mathbf{v} \rangle$. To compute $\frac{\partial F(\mathbf{v})}{\partial \alpha}$:

$$\frac{\partial F(\mathbf{v})}{\partial \alpha} = \sum_{i=1}^{d} \left( \frac{\partial F(\mathbf{v})}{\partial v_i} \frac{\partial v_i}{\partial \alpha} \right) \tag{30}$$

We can substitute value of $\frac{\partial v_i}{\partial \alpha} = (x_i - u_i)$ and $\partial_i F(\mathbf{v})$ from Eq. 29 to Eq. 30.

$$
\begin{aligned}
\frac{\partial F(\mathbf{v})}{\partial \alpha} &= \sum_{i=1}^{d} [w_i H'(z)(x_i - u_i)] \\
&= \langle \mathbf{x} - \mathbf{u}, \mathbf{w} \rangle H'(z)
\end{aligned}
\tag{31}
$$

This gives:

$$dF(\mathbf{v}) = \langle \mathbf{x} - \mathbf{u}, \mathbf{w} \rangle H'(z) \partial \alpha \tag{32}$$

Since $\langle \mathbf{x} - \mathbf{u}, \mathbf{w} \rangle$ is scalar,

$$H'(z) \partial \alpha = \frac{dF(\mathbf{v})}{\langle \mathbf{x} - \mathbf{u}, \mathbf{w} \rangle} \tag{33}$$

Eq. 33 can be used to rewrite the integral in the definition of $IG_i^F(\mathbf{x})$ in Eq. 28,

$$\int_{\alpha=0}^{1} \partial_i F(\mathbf{v}) \partial \alpha = \int_{\alpha=0}^{1} w_i H'(z) \partial z \quad \text{[From Eqn. 29]}$$

$$= \int_{\alpha=0}^{1} w_i \frac{dF(\mathbf{v})}{\langle \mathbf{x} - \mathbf{u}, \mathbf{w} \rangle}$$

$$= \frac{w_i}{\langle \mathbf{x} - \mathbf{u}, \mathbf{w} \rangle} \int_{\alpha=0}^{1} dF(\mathbf{v})$$

$$= \frac{w_i}{\langle \mathbf{x} - \mathbf{u}, \mathbf{w} \rangle} [F(\mathbf{x}) - F(\mathbf{u})] \tag{34}$$

Hence, we obtain the closed form for Integrated Gradients from its definition in Eqn. 28 as

$$IG_i^F(\mathbf{x}, \mathbf{u}) = [F(\mathbf{x}) - F(\mathbf{u})] \frac{(x_i - u_i)w_i}{\langle \mathbf{x} - \mathbf{u}, \mathbf{w} \rangle}$$

$$IG^F(\mathbf{x}, \mathbf{u}) = [F(\mathbf{x}) - F(\mathbf{u})] \frac{(\mathbf{x} - \mathbf{u}) \odot \mathbf{w}}{\langle \mathbf{x} - \mathbf{u}, \mathbf{w} \rangle} \tag{35}$$

Here, $\odot$ is the entry-wise product of two vectors.

Now consider $\mathbf{x}' \in \mathcal{N}_{\mathbf{x}}$ is a noisy version of input image $\mathbf{x}$ where $\mathcal{N}_{\mathbf{x}}$ indicates a neighborhood of inputs $\mathbf{x}$ where the model prediction is locally consistent. The stability of the IG attribution can be computed using Eqn. 36.

$$\Delta = ||IG^F(\mathbf{x}', \mathbf{u}) - IG^F(\mathbf{x}, \mathbf{u})||_1 \tag{36}$$

This is equivalent to,

$$\Delta \approx ||IG^F(\mathbf{x}', \mathbf{x})||_1$$

$$= \left\| [F(\mathbf{x}') - F(\mathbf{x})] \frac{(\mathbf{x}' - \mathbf{x}) \odot \mathbf{w}}{\langle \mathbf{x}' - \mathbf{x}, \mathbf{w} \rangle} \right\|_1$$

$$= \left\| [F(\mathbf{x}') - F(\mathbf{x})] \frac{\Delta_x \odot \mathbf{w}}{\langle \Delta_x, \mathbf{w} \rangle} \right\|_1 \tag{37}$$

Assuming $\mathbf{w}$ to be constant for a given model, we can conclude from Eqn. 37 that the sensitivity of the IG attribution is a direct result of the sensitivity of the model $||F(\mathbf{x}') - F(\mathbf{x})||$.

### H.3   Relationship for SmoothGrad (SG) (Smilkov et al., 2017)

To compute SmoothGrad (SG), we introduce Gaussian noise $\mathbf{n} \sim \mathcal{N}(0, \sigma^2)$ to the input $\mathbf{x}$ and compute the input-gradient for multiple noisy samples $\mathbf{x}_k = \mathbf{x} + \mathbf{n}_k$ for $k = 1, \ldots, N$, where $N$ is the number of noise samples.

$$SG(\mathbf{x}) = \frac{1}{N} \sum_{k=1}^{N} \frac{\partial F(\mathbf{x}_k)}{\partial \mathbf{x}_k} \tag{38}$$

SG explanation is then obtained by averaging the explanations. Since SG is a simple averaging of Vanilla Gradient, the relationship for SG follows from relationship of VG, as shown in Section H.1.

# I   Evaluation metrics

Below, we discuss evaluation metrics used in our experiments.

## I.1   Sparsity (Chalasani et al., 2020)

We measure the sparsity of the attribution vector $\phi(\mathbf{x})$ by computing its Gini index. Given a vector of attribution $\phi(\mathbf{x}) \in R^d$, the absolute of the vector is first sorted in non-decreasing order, and the Gini index is computed using Eqn. 39.

$$G(\phi(\mathbf{x})) = 1 - 2 \sum_{k=1}^{d} \frac{\phi(\mathbf{x})_{(k)}}{||\phi(\mathbf{x})||_1} \frac{d - k + 0.5}{d} \tag{39}$$

The formula calculates a weighted sum of fractions, where each fraction represents the contribution of the k-th largest element to the overall sparsity. The formula assigns greater weight to larger elements and smaller weight to smaller elements. The Gini Index values lie in between $[0, 1]$; A value of 1 indicates perfect sparsity, where only one element in the vector $\phi_i(\mathbf{x}) > 0$. The sparsity is zero if all the vectors are equal to some positive value.

## I.2   Stability (Agarwal et al., 2022)

The stability metric measures how similar explanations are for similar inputs. Relative input stability (given by Eqn. 40) is measured as the difference between two attribution vectors $\phi(\mathbf{x})$ and $\phi(\mathbf{x}')$ with respect to the difference between the two inputs $\mathbf{x}$ and $\mathbf{x}'$. $\mathbf{x}'$ is computed by perturbing $\mathbf{x}$. A lower RIS value shows that explanations are similar for similar inputs.

$$RIS = max_{\mathbf{x}'} \frac{||\frac{\phi(\mathbf{x}) - \phi(\mathbf{x}')}{\phi(\mathbf{x})}||}{max(||\frac{\mathbf{x} - \mathbf{x}'}{\mathbf{x}}||_p, \epsilon_{min})} \tag{40}$$
$$\forall \mathbf{x}' \; s.t. \; \mathbf{x}' \in \mathcal{N}_{\mathbf{x}}; \hat{y}_{\mathbf{x}} = \hat{y}_{\mathbf{x}'}$$

Relative input stability only measures the difference in input space and does not measure whether there was a change in the logic path of a network for a perturbed input. Relative representation stability (given by Eqn. 41) uses the internal representation of the model $(a(.))$ to compute the stability.

$$RRS = max_{\mathbf{x}'} \frac{||\frac{\phi(\mathbf{x}) - \phi(\mathbf{x}')}{\phi(\mathbf{x})}||}{max(||a(\mathbf{x}) - a(\mathbf{x}')||_p, \epsilon_{min})} \tag{41}$$
$$\forall \mathbf{x}' \; s.t. \; \mathbf{x}' \in \mathcal{N}_{\mathbf{x}}; \hat{y}_{\mathbf{x}} = \hat{y}_{\mathbf{x}'}$$

Relative output stability (given by Eqn. 42) measures the difference between two attribution vectors $\phi(\mathbf{x})$ and $\phi(\mathbf{x}')$ with respect to the difference between the model logits for two inputs $z(\mathbf{x})$ and $z(\mathbf{x}')$ when $\mathbf{x}$ is perturbed to produce $\mathbf{x}'$. A lower ROS value shows that explanations are similar for similar inputs.

$$ROS = max_{\mathbf{x}'} \frac{||\frac{\phi(\mathbf{x}) - \phi(\mathbf{x}')}{\phi(\mathbf{x})}||}{max(||z(\mathbf{x}) - z(\mathbf{x}')||_p, \epsilon_{min})} \tag{42}$$
$$\forall \mathbf{x}' \; s.t. \; \mathbf{x}' \in \mathcal{N}_{\mathbf{x}}; \hat{y}_{\mathbf{x}} = \hat{y}_{\mathbf{x}'}$$

$\mathcal{N}_{\mathbf{x}}$ in Eqn. 40, Eqn. 41 and Eqn. 42 indicates a neighborhood of inputs $\mathbf{x}'$ similar to $\mathbf{x}$. We use the implementation of the stability metrics available in Quantus (Hedström et al., 2023).

### I.3 ROAD: Remove and Debias (Rong et al., 2022)

Faithfulness metrics that involve pixel removal and measuring model prediction changes (such as insertion/deletion (Petsiuk et al., 2018)) introduces artifacts and cause a distribution shift in the perturbed inputs. Retraining based approaches like ROAR (Hooker et al., 2019) addresses this problem but is computationally expensive. ROAD (Rong et al., 2022) addresses both concerns in faithfulness evaluation.

ROAD measures the accuracy of a model on the provided test set at each step of an iterative process of removing $k$ most important pixels. Removal of pixels is done with a noisy linear imputation to avoid out-of-distribution samples. We set $k = 5$ in our experiments, and adopt the MoRF (Most Relevant First) removal strategy where a faster drop in accuracy with increase in removal of $k$ most important features indicate that key discriminative features are being removed. ROAD demonstrates consistent results with both MoRF and LeRF (Least Removal First) removal strategy. For further details, see Rong et al. (2022).

### I.4 Structural similarity (Adebayo et al., 2018)

Structural similarity measures the structural similarity between saliency maps of original and perturbed samples, given the same model prediction. We measure the similarity of saliency maps using the structural similarity index (SSIM). For each image, we add Gaussian noise and generate its noisy version such that the model prediction is consistent. We then compute the saliency map of the two images and measure the structural similarity between the maps.

## J  Additional Visualization

We provide additional visualizations on Vanilla Gradient (VG) in Figures 17, 18 and 19 for various models: naturally-trained (N), adversarially-trained (A), adversarial training with mean-filter smoothing (M1), adversarial training with median-filter smoothing (M2), adversarial training with Gaussian-filter smoothing (G), adversarial training with embedded filter smoothing (E), and adversarial training with non-local gaussian smoothing (NG). We can observe that saliency maps from the adversarial models (A) are sparser than the naturally trained model (N). Adversarially trained models with local feature map smoothed models (M1, M2, G) reduce the sparsity to improve stability. The use of non-local smoothing filters (E and NG) increases the sparsity further.

We plot the saliency maps using Integrated Gradients (IG) for various models in Figures 20, 21 and 22. As illustrated, IG produces more fine-grained saliency maps than Vanilla Gradient even with a naturally trained model. Robust models increase the sparsity of such saliency maps, compromising stability. Adding local filters like median during adversarial training reduces sparsity to enhance stability.

We provide illustrations for SmoothGrad (SG) in Figures 23, 24 and 25 where we can observe that saliency maps of naturally trained models are visually sharper and coherent because of averaging. However, using robust models increases the sparsity and produces more comprehensible saliency maps.

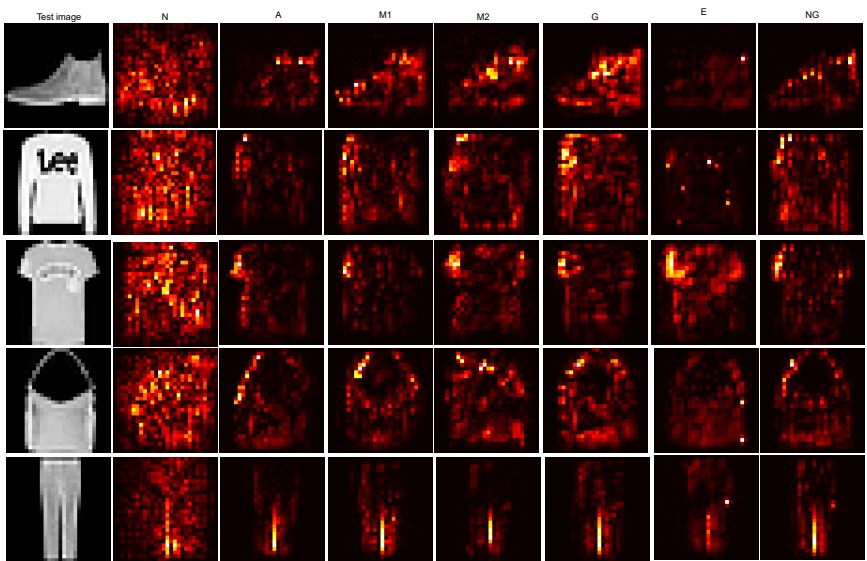

Figure 17: Additional visualization for VG (FMNIST) (N: naturally-trained, A: adversarially-trained, M1: adversarially-trained with mean-filter, M2: adversarially-trained with median-filter, G: adversarially-trained with Gaussian-filter, E: adversarially-trained with embedded filter, NG: adversarially-trained with non-local gaussian)

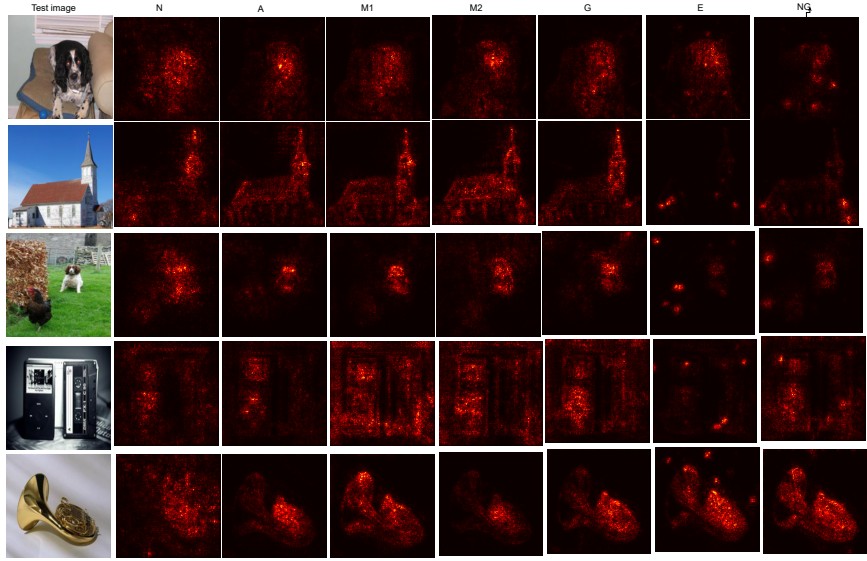

Figure 18: Additional visualization for VG (ImageNette) (N: naturally-trained, A: adversarially-trained, M1: adversarially-trained with mean-filter, M2: adversarially-trained with median-filter, G: adversarially-trained with Gaussian-filter, E: adversarially-trained with embedded filter, NG: adversarially-trained with non-local gaussian)

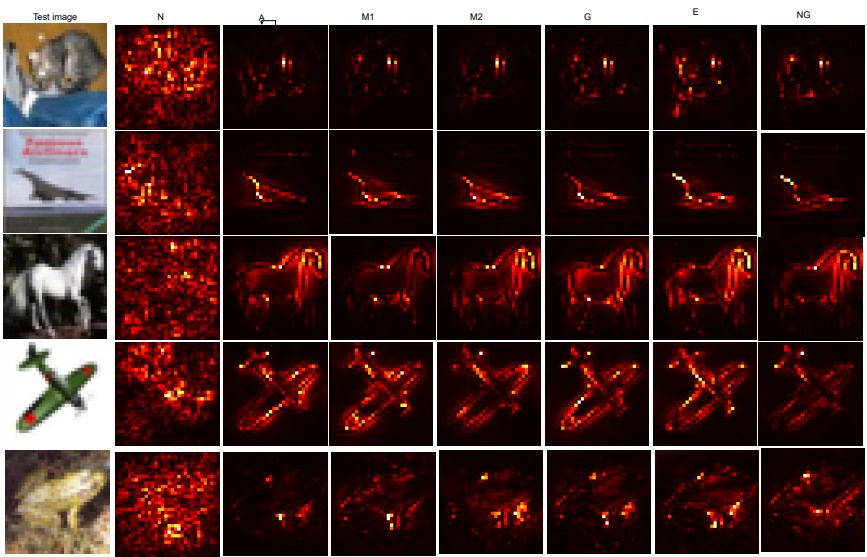

Figure 19: Additional visualization for VG (CIFAR-10) (N: naturally-trained, A: adversarially-trained, M1: adversarially-trained with mean-filter, M2: adversarially-trained with median-filter, G: adversarially-trained with Gaussian-filter, E: adversarially-trained with embedded filter, NG: adversarially-trained with non-local gaussian)

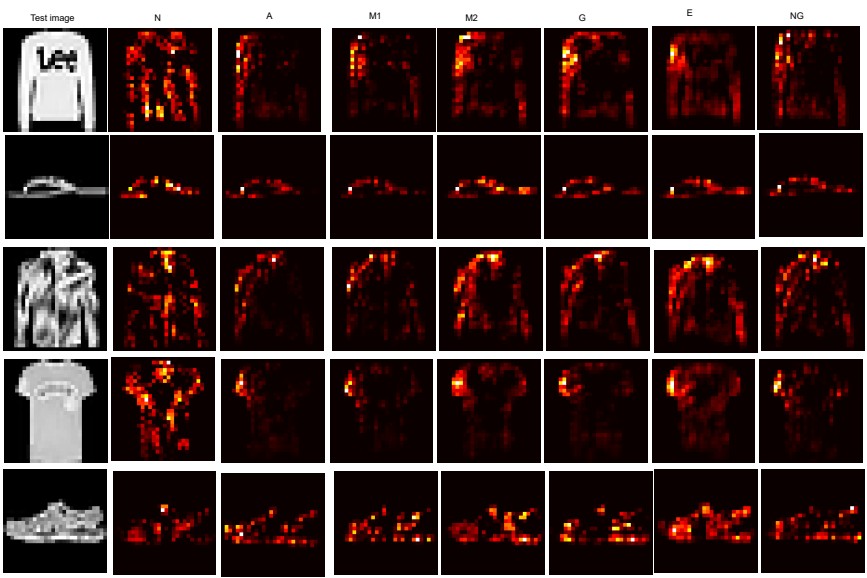

Figure 20: Saliency maps visualization on FMNIST using IG across different models (N: naturally-trained, A: adversarially-trained, M1: adversarially-trained with mean-filter, M2: adversarially-trained with median-filter, G: adversarially-trained with Gaussian-filter, E: adversarially-trained with embedded filter, NG: adversarially-trained with non-local gaussian).

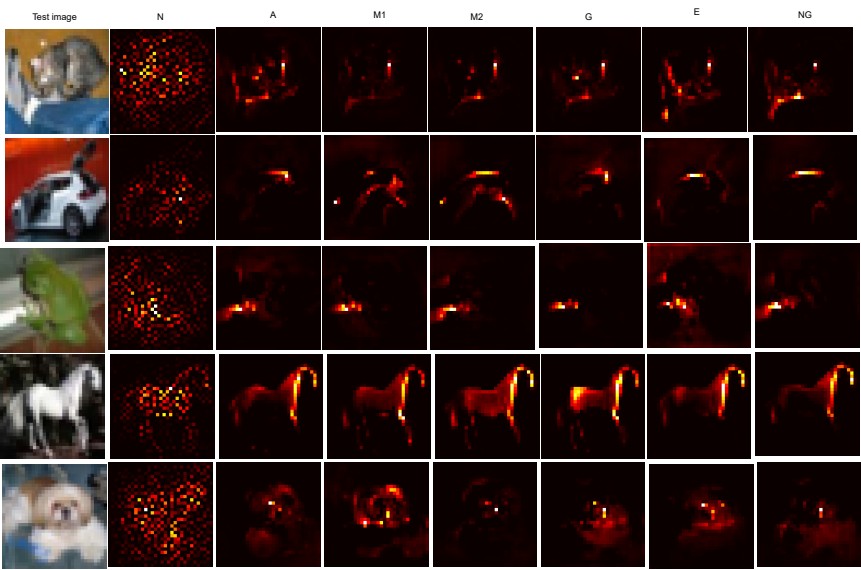

Figure 21: Saliency maps visualization on CIFAR-10 using IG across different models (N: naturally-trained, A: adversarially-trained, M1: adversarially-trained with mean-filter, M2: adversarially-trained with median-filter, G: adversarially-trained with Gaussian-filter, E: adversarially-trained with embedded filter, NG: adversarially-trained with non-local gaussian).

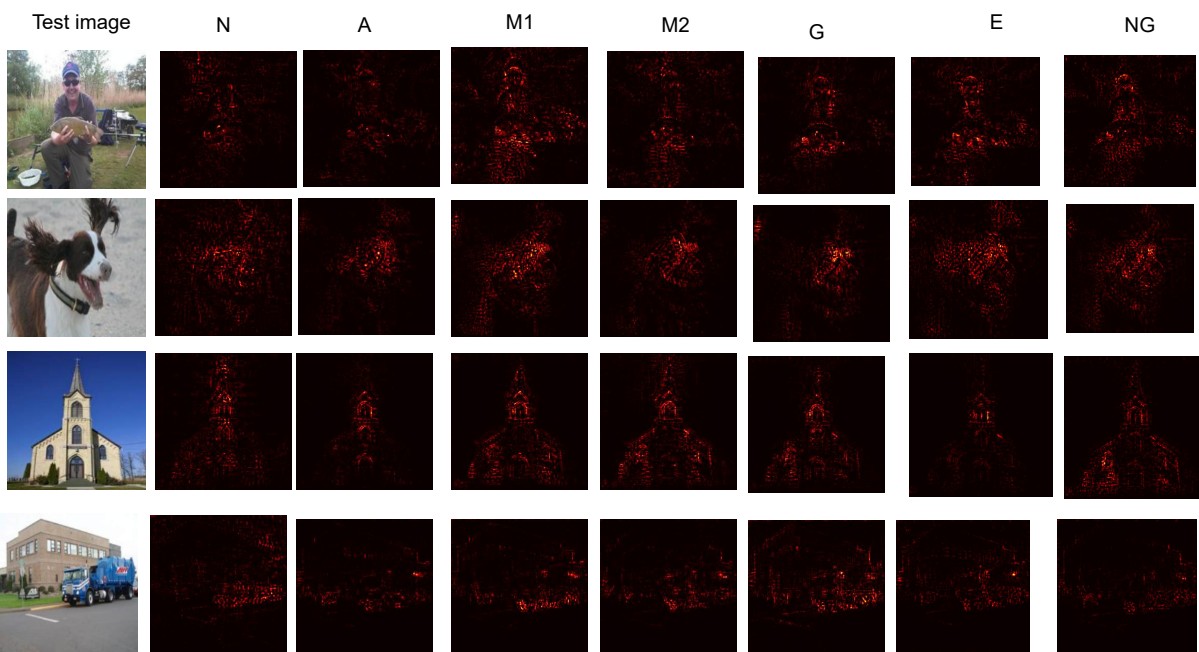

Figure 22: Saliency maps visualization on ImageNette using IG across different models (N: naturally-trained, A: adversarially-trained, M1: adversarially-trained with mean-filter, M2: adversarially-trained with median-filter, G: adversarially-trained with Gaussian-filter, E: adversarially-trained with embedded filter, NG: adversarially-trained with non-local gaussian).

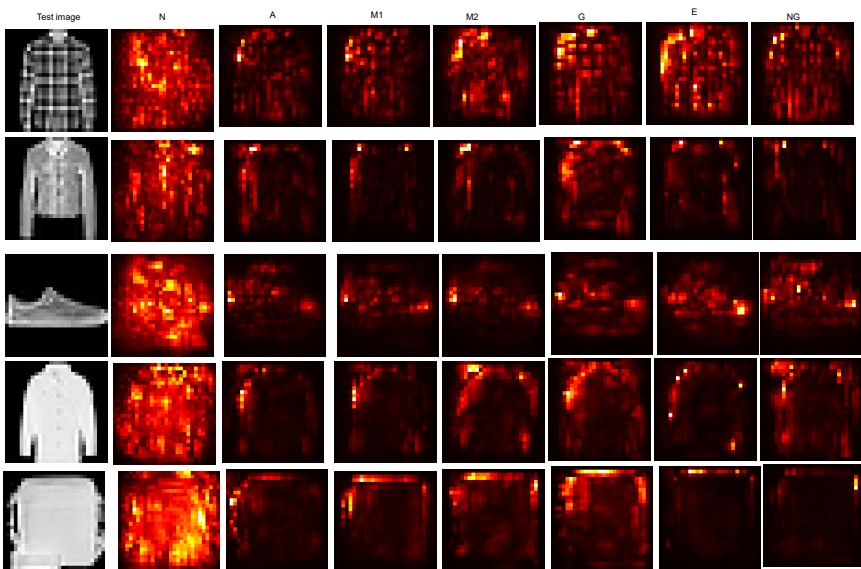

Figure 23: Saliency maps visualization on FMNIST using SmoothGrad across different models (N: naturally-trained, A: adversarially-trained, M1: adversarially-trained with mean-filter, M2: adversarially-trained with median-filter, G: adversarially-trained with Gaussian-filter, E: adversarially-trained with embedded filter, NG: adversarially-trained with non-local gaussian).

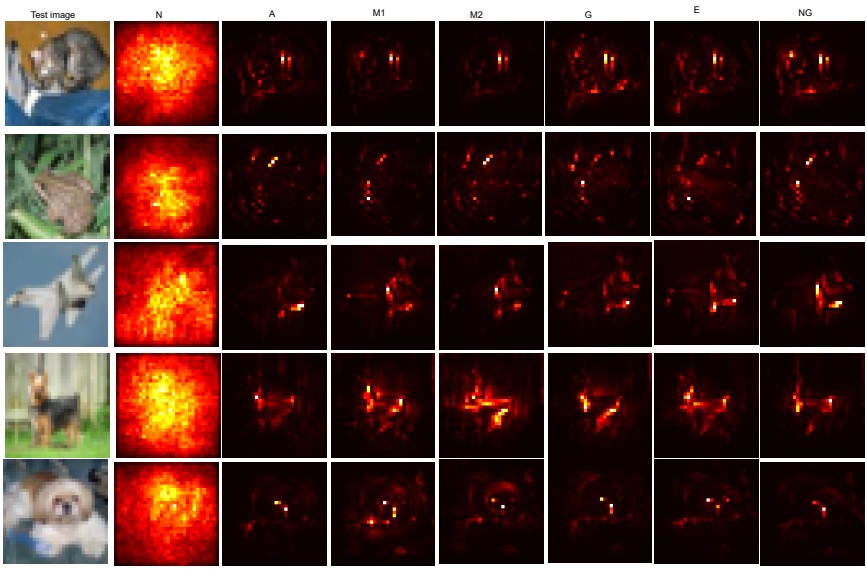

Figure 24: Saliency maps visualization on CIFAR-10 using SmoothGrad across different models (N: naturally-trained, A: adversarially-trained, M1: adversarially-trained with mean-filter, M2: adversarially-trained with median-filter, G: adversarially-trained with Gaussian-filter, E: adversarially-trained with embedded filter, NG: adversarially-trained with non-local gaussian).

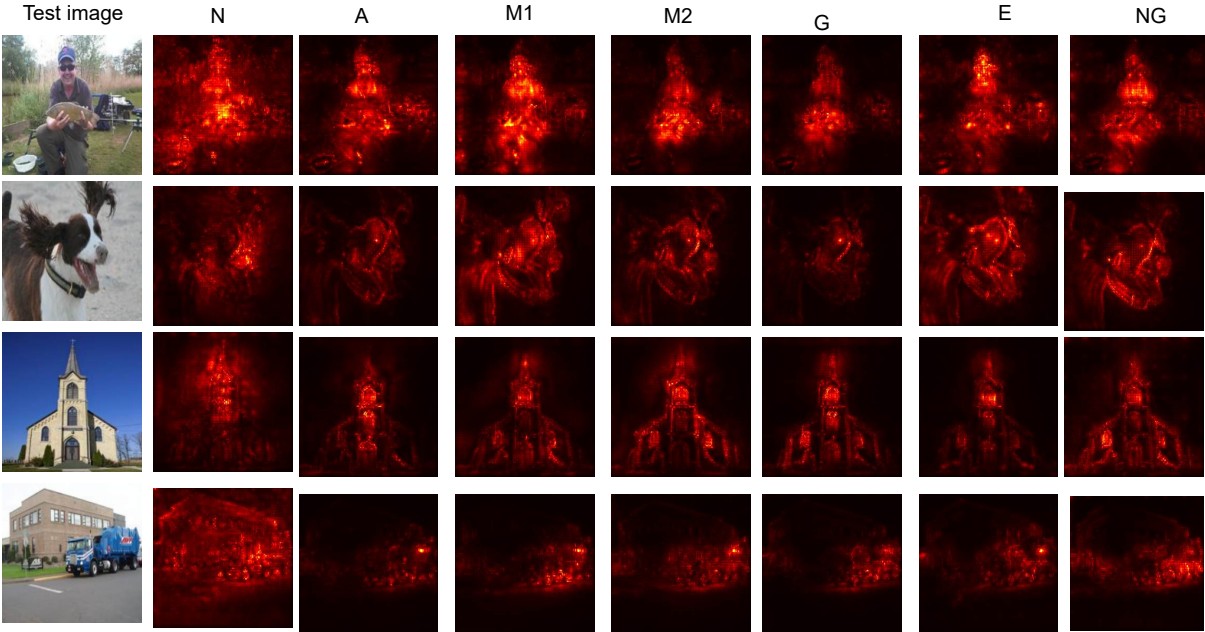

Figure 25: Saliency maps visualization on ImageNette using SmoothGrad across different models (N: naturally-trained, A: adversarially-trained, M1: adversarially-trained with mean-filter, M2: adversarially-trained with median-filter, G: adversarially-trained with Gaussian-filter, E: adversarially-trained with embedded filter, NG: adversarially-trained with non-local gaussian).

