# OpenReview forum: "Towards improving saliency map interpretability using feature map smoothing"
_TMLR — Rejected by TMLR_

### Review · Reviewer_yDKJ · 2025-02-26

**Summary Of Contributions:**

This work explores the relationship between model robustness and the explainability of attribution methods. It theoretically and empirically demonstrates that a model’s robustness to perturbations influences the explainability of its attributions. The authors investigate the effects of different smoothing techniques combined with adversarial training, showing that the reconstructed models result in enhanced explainability of attributions. The experiments evaluate three attribution methods using various explanation evaluation metrics to support the proposed approach.

**Audience:**

Yes

**Broader Impact Concerns:**

There are no additional ethical concerns identified that would require further elaboration.

**Claims And Evidence:**

Yes

**Requested Changes:**

1. The authors are suggseted to adjust their motivations and evaluation targets to create a more cohesive and self-consistent claim.
2. Some important details should be included in the main paper rather than appendix in order to make the paper more self-contained. For instance, the definition and description of the evaluated filter should be formally introduced in Section 3.3. Additionally, the results discussed in Section 4.3 could be more effectively presented in a table for clarity and ease of comparison.
3. The performance of models trained under different robustness settings should be explicitly reported to give a more comprehensive view of their performance.
4. The qualitative analysis in Sec. 4.3 lacks rigor. The evaluation based on human perception (e.g., testing whether students find explanations more understandable) does not faithfully assess whether the attributions accurately reflect the model’s reasoning. The target of explanation methods should be to faithfully represent the model’s behavior rather than generate explanations that are easy for humans to understand. The authors are suggested to highlight the motivations of including this experiment.
5. The abbreviations used for the compared models are too concise, which may hinder readers’ ability to quickly understand their meaning. Using more descriptive names for the models would improve clarity and facilitate easier understanding.
6. Figure captions should provide more detailed context rather than relying on short titles. More informative captions would enhance readability and allow the figures to stand alone. Additionally, figures should be placed closer to their references in the text (e.g., Figs. 4–6).

**Strengths And Weaknesses:**

**Strengths**:
1. The paper is well-organized and presented in a clear and accessible manner.
2. The study provides valuable insights into the relation between robustness and attribution explainability, contributing to a better understanding of explainability in attributions.
3. The experimental results are well-structured and accompanied by reasonable interpretations.

**Major Weaknesses**:

The primary issue with this work lies in the mismatch between the motivations and the methods. While the authors claim to offer a method for generating more reliable explanations, the focus of the paper is primarily on enhancing the robustness of models, rather than improving the consistency or faithfulness of attribution methods. If the primary goal is to explore the effect of robustness on explainability, the authors should refine their focus to better demonstrate how model robustness translates into more faithful and consistent explanations. If the intention is to construct more robust models, the paper would benefit from a more direct comparison of different robustness techniques to highlight the advantages of the proposed method. As it stands, the authors should adjust their motivations and evaluation targets to create a more cohesive and self-consistent claim.

**Minor Weaknessnes**:

Please see below the requested changes for more minor weaknesses.

---

> ### Author Response · Authors · 2025-03-21
> **Response to Reviewer yDKJ**
>
> We sincerely appreciate the reviewer for their insightful comments and suggestions. We have made the requested changes in the updated pdf. All the changes are highlighted in blue. Response to suggested changes:
>
> - **Adjust their motivations**: We have now explicitly mentioned in the Introduction paragraph that we take a complementary approach to improving the quality of saliency maps by studying their characteristics in naturally and adversarially trained models.
>
> - **Include important details**: As requested by the reviewer, we have formally defined the filters used in this work in Section 3.3.
> - **Results in a table & highlight motivation**: The statistical tests were performed to determine whether the observed differences in participant ratings across the three models were statistically significant. Without such an analysis, it would be unclear whether the trends observed in the mean ratings were due to actual effects of the training strategies or merely due to chance. We have clarified the motivation by adding more details, also added the table for clarity, and more details which were in the Appendix before.
> - **Performance of models trained under different robustness settings**: For the models used in Section 4.1, we have presented the model performance in Appendix C.
> - **More descriptive names for the models**: We have replaced the acronyms used in the paper in Tables, Figures and analysis with full names to improve clarity.
> - **On figure captions and location**: We have fixed the figure captions, updated the analysis by dataset and then placed figures closer to their reference in the text

---

> > ### Comment · Reviewer_yDKJ · 2025-03-30
> > **Concerns on Methodological Scope and Rigor**
> >
> > I appreciate the authors' response and revisions. However, my primary concerns remain unresolved. The authors claim their goal is "improving the quality of saliency maps by studying their characteristics across different models." However, this statement is somewhat confusing. The improvement of saliency map quality and the identification of characteristics of explanation tools across models are not inherently linked. More robust models do not lead to better reliability and explanation quality of an attribution method.
> >
> > From my understanding, the authors are primarily focused on exploring the characteristics of explanation tools across various models. However, limiting the study to adversarially trained models significantly narrows the impact of this work. Additionally, the attribution methods discussed, including InputGrad, IG, and SG, are not sufficient to draw comprehensive conclusions.
> >
> > Overall, the authors seem to implicitly present the improvement of explanation quality as a key contribution, but this lacks rigor in the context of their work. Given that the primary contribution appears to be the robust technique, I still have concerns regarding the limited comparison of explanation methods (i.e., InputGrad, IG, and SG) and the narrow focus on adversarially trained models.

---

> > > ### Author Response · Authors · 2025-03-31
> > > **On scope and rigor of the work**
> > >
> > > We thank the reviewer for their continued engagement and the thoughtful critique.
> > >
> > > Our primary goal is not to claim that robustness automatically improves explanation quality, but first identify the relationship between input-gradient based explanation methods and model sensitivity and investigate how model training techniques—particularly adversarial training and its extensions—influence the quality of saliency maps in such input-gradient based attribution methods. In that sense, while robustness and explanation quality are not inherently linked, we demonstrate that certain training-time modifications (like feature-map smoothing) can encourage better trade-offs among explanation desiderata.
> > >
> > > We also agree that attribution methods vary widely in their underlying principles and behavior. Our focus on VG, IG, and SG stems from their shared reliance on input gradients, which is central to our hypothesis about the interplay between model sensitivity and explanation quality. While these methods don’t cover the full landscape of explainability techniques, they are representative of a widely used family of attribution methods.
> > >
> > > To clarify this scope, we can update the discussion section to acknowledge the limitations of our study, and propose future work that would extend this framework to perturbation-based methods.

---

> > > ### Author Response · Authors · 2025-04-10
> > > **Updated changes**
> > >
> > > We thank the reviewer again for their insightful comments. We have expanded the limitations and included the current work focusing only on input-gradient based methods. We have also softened the claim of faithfulness, focusing on the contributions that are empirically supported.

---

### Review · Reviewer_iEtN · 2025-03-05

**Summary Of Contributions:**

This paper explores the trade-off between stability and sparsity in input-gradient-based saliency maps for explaining image classifiers. The authors find that while adversarial training enhances robustness and produces sparser explanations, it often reduces stability. To address this, the authors introduce a smoothing layer during adversarial training, improving stability and faithfulness without sacrificing sparsity. A complementary user study confirms that human evaluators find these explanations more interpretable and trustworthy.

**Audience:**

Yes

**Broader Impact Concerns:**

N/A.

**Claims And Evidence:**

No

**Requested Changes:**

- Improve presentation of Figures 4-6 and Figures 7-9, it was difficult for me as a reader to compare across them.
- Fix inconsistent Numbering: Section 4.2.3 seems to be skipped..
- Clearly define which models you refer to when “robust models” are mentioned. Is it {A, M,  M2, G, E, NG} or a subset thereof?
- Clarify why “robust accuracy” is selected in Figs. 12 and 13 over “accuracy” Figs. 10 and 11.
- Clarify notation in Section 3.1, ensuring consistency in the definitions in Section 3.1 and use standard notation.
- Improve discussion on the distinction between model robustness and explanation method robustness.
- Include a random baseline (random saliency map) in Table 1 and all ROAD evaluations.
- Add dROAD score in Table 1 using "area under the perturbation curves" (AUPC).
- Justify the claim that robust models surpass natural models in accuracy drop, particularly for IG in Figure 4a.
- Provide clearer evidence that smoothing filters do not affect explanation faithfulness.
- Define "estimated faithfulness" and clarify its relation to ROAD faithfulness scores in Figures 4-6.
- Add references to Figures, Tables, or Sections to substantiate claims in Section 4.2.5.
- Ensure clear support for the claim that "saliency maps in robust models are more faithful."
Include discussion and related works on the tradeoff between sparsity-faithfulness-robustness, e.g.
Trade-off between faithfulness and sparsity, where less sparse (higher entropy) attention vectors are less faithful to model predictions [Eber22].
- Clarify where the smoothing layers should be applied.

*References*

[Szeg13] Szegedy, C., Zaremba, W., Sutskever, I., Bruna, J., Erhan, D., Goodfellow, I., & Fergus, R. (2013). Intriguing properties of neural networks. arXiv preprint arXiv:1312.6199.

[Mont14] Montufar, G. F., Pascanu, R., Cho, K., & Bengio, Y. (2014). On the number of linear regions of deep neural networks. Advances in neural information processing systems, 27.

[Bald17] Balduzzi, D., Frean, M., Leary, L., Lewis, J. P., Ma, K. W. D., & McWilliams, B. (2017, July). The shattered gradients problem: If resnets are the answer, then what is the question?. In International conference on machine learning (pp. 342-350). PMLR.

[Eber22] Eberle, O., Brandl, S., Pilot, J., & Søgaard, A. (2022, May). Do transformer models show similar attention patterns to task-specific human gaze?. In Proceedings of the 60th Annual Meeting of the Association for Computational Linguistics (Volume 1: Long Papers) (pp. 4295-4309).

**Strengths And Weaknesses:**

*Strengths*
- The paper is overall well-written, easy to follow and addresses the assessment of the robustness-faithfulness-accuracy tradeoffs.

- Extensive evaluation on three datasets, three explanations methods and various evaluation metrics.

- Qualitative Analysis of saliency map quality to assess if users prefer adversarial models over naturally trained models.

- Code is provided and the experiments appear reproducible.


*Weaknesses*

- Equations in Section 3.1 formulate the model function as an activation function, e.g. sigmoid, that is dependent on $(w,x)$, which appears to be  an arbitrary choice. The introduction of $H$ is also not consistent across methods (VG, IG), and does not reappear in the main equations used to motivate the bounds (Eqs. 5-7). The use of $.w$ is not common notation.

- It is well-known that input gradients are not robust for deep models [Szeg13, Mon14, Bald17], which raises the question if this is the case due to the model or the used explanation method. It would be important to discuss this distinction more clearly in the paper.

- Change of the explanation methods (Figure 4 vs Figure 5 vs Figure 6), seems to have a strong effect on the ranking of different training/smoothing variants investigated. Overall, it would be good to include a random baseline (randomly sampled saliency map) for the evaluation in addition to VG, IG, SG for Table 1 and also for all ROAD evaluations. For easier interpretation and evaluation of results, it would be very helpful to include a dROAD score in Table 1 using the common “area under the perturbation curves” (AUPC) score to better understand the relation between sparsity, stability and faithfulness.

- The following statements made in Section 4.2.2. are not sufficiently supported and were not visible for me in the associated Figures:

    - “Across VG,IG and SG,on FMNIST(See Figure 4a, 5a, 6a), while natural models start with sharper drop in accuracy, robust models quickly surpass them.”  This appears to only apply to IG in Figure 4a.
    - “The results confirm that saliency maps from robust models are consistently more faithful compared to those from naturally trained models, and the introduction of smoothing filters does not affect explanation faithfulness.” I don’t see this clearly as a result in the presented data.

- How is “estimated faithfulness” computed? I could not find a definition in the provided reference (Alvarez Melis & Jaakkola, 2018). How does this metric relate to the ROAD faithfulness scores presented in Figs. 4-6?

- In Section 4.2.5, it is not clear where the claims a-e are supported, please add evidence that support these, e.g. by providing cross-refs to Figures, Tables or Sections. For example, “(e) saliency maps in robust models are more faithful to the underlying model than naturally trained counterparts.”, it is difficult to check this or reconstruct for the reader where this statement is supported.


*References*

[Szeg13] Szegedy, C., Zaremba, W., Sutskever, I., Bruna, J., Erhan, D., Goodfellow, I., & Fergus, R. (2013). Intriguing properties of neural networks. arXiv preprint arXiv:1312.6199.

[Mont14] Montufar, G. F., Pascanu, R., Cho, K., & Bengio, Y. (2014). On the number of linear regions of deep neural networks. Advances in neural information processing systems, 27.

[Bald17] Balduzzi, D., Frean, M., Leary, L., Lewis, J. P., Ma, K. W. D., & McWilliams, B. (2017, July). The shattered gradients problem: If resnets are the answer, then what is the question?. In International conference on machine learning (pp. 342-350). PMLR.

---

> ### Author Response · Authors · 2025-03-21
> **Response to Reviewer iEtN**
>
> We sincerely appreciate the reviewer for their insightful comments and suggestions. We have made the suggested changes in the updated paper, highlighted in blue. Response to suggested changes:
>
> - **Improve presentation of Figures**: We have combined figures (e.g., Figures 4/5/6, 7/8/9) by dataset and then rewritten the analyses to improve the presentation.
> - **Fix inconsistent Numbering**: We have fixed the inconsistencies.
> - **Clearly define which models you refer to when “robust models” are mentioned**: In the earlier version, robust models referred to models trained with adversarial training with or without smoothing block. However, we agree this might cause confusion. So, we have replaced the usage of robust models with specific model training methods used in our study. We want to thank the reviewer for pointing this out.
> - **Clarify why “robust accuracy” is selected in Figs. 12,13**: In Figures 10,11, we plot the natural accuracy vs saliency map quality for adversarially trained models trained with increasing adversarial perturbation strength. It shows that the adversarially trained models (with feature map smoothing) with higher perturbation strength have lower natural accuracy but better saliency map quality. In Figures 12 & 13, we show that consequently, while the benign accuracy for such models reduces as the adversarial perturbation strength is increased, robust accuracy improves. Figures 12 & 13 show that with the increase in robust accuracy, saliency map quality improves. We have clarified this in Section 4.2.5.
> - **On Section 3.1**: The primary reason for using a single-layer neural network is to derive closed-form expressions for gradient-based attributions. For deeper networks, the expressions for these attribution methods involve complex compositions of nonlinear transformations and weight matrices, making it infeasible to obtain exact analytical form. The single-layer formulation aligns with prior theoretical work in explainability such as \citet{chalasani2020concise}, which provides closed-form IG expressions under linear model assumptions. We understand this can be considered a limitation of our study so we describe this in our limitation section. As suggested by the reviewer, we have also reformulated the definition of VG(x) and SG(x) with F(x) so that the bounds are now consistent, and the mistake of representing gradient with .w is also edited. We want to thank the reviewer for pointing these out.
> - **Improve discussion on the distinction between model robustness & explanation method robustness**: Model robustness is the model performance in detecting (correctly) adversarially perturbed samples, whereas explanation method robustness is maintaining similar saliency maps, in terms of stability and structural similarity, on perturbed samples. We have clarified this in Section 4.2.5
> - **Define "estimated faithfulness**: Faithfulness estimate measures faithfulness of explanations as a correlation score by iteratively modifying a given image and computing the correlation between the sum of attributions and the difference in model prediction. In comparison to the visual analysis of ROAD, this provides a direct quantitative evaluation of saliency maps. We have now clarified this in Section 4.2.4. This metric is defined in the following XAI toolbox https://quantus.readthedocs.io/en/latest/docs_api/quantus.metrics.faithfulness.faithfulness_estimate.html
> - **Add references to claims in Section 4.2.5**: We have added references.
> - **Ensure clear support for the claim that "saliency maps in robust models are more faithful."**: We have added related works from \citet{shah2021input} which showed that naturally trained models fail to capture the most discriminative features, often due to feature leakage and \citet{eberle2022transformer} which showed that less sparse attention vectors in transformers are less faithful to the model predictions, to support the claim in Section 4.2.4.
> - **Clarify where the smoothing layers should be applied:** Smoothing layers are applied after the first convolutional block or residual block. We describe the methodology in Section 4.1 with experiments in Appendix D.

---

> > ### Author Response · Authors · 2025-03-21
> > **Response to Reviewer iEtN**
> >
> > - **Include a random baseline in Table 1 and all ROAD**: Random saliency map can be an excellent baseline for comparing ROAD evaluation. We really want to thank the reviewer for bringing this up. We have performed ROAD evaluations for random saliency maps for each dataset and included them in the paper along with dROAD score in Table 2.
> >
> > However, we want to point out that the stability metrics used in Table 1 do not provide any information on the stability of random saliency maps. To clarify this further, let's consider computing the input stability of an explanation method, which measures how much the saliency map changes when the input image is perturbed (e.g., with Gaussian noise) ensuring the model prediction remains unchanged. In the case of a random saliency method, the generated saliency maps for the original and perturbed images are purely random and do not correspond to the model’s sensitivity. As a result, the stability metric does not provide insight into the effect of perturbations. Instead, it only reflects the randomness in the generation process. To address the reviewer’s concern, we computed stability metrics for the random method and present the results below. As expected, the stability values for the random method do not exhibit significant trends compared to other explanation methods in Section 4.2.1.
> >
> > In the table, RG refers to random gradient method. A: adversarially trained model, M1: adversarially trained with mean filter, with median filter (M2), with Gaussian filter (G), with embedded filter (E), with nonlocal Gaussian filter (NG). dG, dRIS, dROS, dRRS measure change in sparsity, input stability, output stability and representation stability with respect to naturally trained model.
> >
> > | Dataset           | Metric  | A      | M1     | M2     | G      | E      | NG     |
> > |------------------|---------|--------|--------|--------|--------|--------|--------|
> > |                  | dG      | 0.000  | 0.000  | 0.000  | 0.000  | 0.000  | 0.000  |
> > | **FMNIST RG**     | dRIS    | -0.256 | -0.202 | -0.248 | -0.222 | 0.092  | -0.246 |
> > |                  | dROS    | 0.101  | -0.031 | 0.113  | 0.048  | 0.206  | 0.114  |
> > |                  | dRRS    | 0.177  | 0.206  | 0.977  | 0.884  | 0.084  | 0.250  |
> > |                  |  |  |  | |   |  |   |
> > |                  | dG      | 0.000  | 0.000  | 0.000  | 0.000  | 0.000  | 0.000  |
> > | **CIFAR RG**      | dRIS    | -0.178 | -0.103 | 0.126  | 0.042  | 0.231  | -0.169 |
> > |                  | dROS    | 1.311  | 0.557  | 0.563  | 0.674  | 0.723  | 0.626  |
> > |                  | dRRS    | 0.824  | 0.844  | 0.890  | 1.110  | 0.753  | 0.701  |
> > |                  |  |  |  | |   |  |   |
> > |                  | dG      | -0.003 | 0.000  | 0.000  | 0.000  | 0.000  | 0.000  |
> > | **ImageNette RG** | dRIS    | -0.399 | -0.139 | -0.030 | -0.254 | -0.348 | -0.437 |
> > |                  | dROS    | -0.524 | 0.752  | -0.396 | -0.579 | -0.615 | -0.421 |
> > |                  | dRRS    | -0.271 | -0.115 | -0.366 | -0.341 | -0.179 | -0.537 |
> >
> >
> > - **Add dROAD score in Table 1 using "area under the perturbation curves" (AUPC)**: We have computed dROAD score as area under the perturbation curve (Table 2) that now supplements the observation from the ROAD plots. We again want to thank the reviewer for suggesting this as this score now quantitatively validates the observations from the ROAD plot.
> >
> > - **Justify the claim that robust models surpass natural models in accuracy drop, in ROAD figures**: We have modified Section 4.2.2 on faithfulness by comparing the ROAD plot and the corresponding dROAD in Table 2.
> >
> > - **Provide clearer evidence that smoothing filters do not affect explanation faithfulness:** The dROAD score (Table 2), computed as the reviewer’s suggestion, quantitatively demonstrates how smoothing filters help to improve explanation faithfulness. In FMNIST, including Gaussian smoothing improves the faithfulness of Vanilla Gradient, and Embedded smoothing improves the faithfulness of Integrated Gradient and SmoothGrad. Similarly, in CIFAR, including any smoothing filters with adversarial training improves the faithfulness of Vanilla Gradient and Integrated Gradient. For SmoothGrad, in CIFAR, except the mean filter, including all other smoothing filters, improve the faithfulness. In ImageNette, all smoothing filters improve faithfulness for Vanilla Gradient. For Integrated Gradient, Gaussian, and Embedded filters improve faithfulness. SmoothGrad for ImageNet is the only approach where adversarial training without smoothing has the highest dROAD score with a median filter marginally close. We have modified our analysis such that it captures both the ROAD plot and the dROAD table.

---

> > > ### Author Response · Authors · 2025-03-21
> > > **Response to Reviewer iEtN**
> > >
> > > The reviewer also asked whether the robustness of the explanation is due to the method or the model **"It is well-known that input gradients are not robust for deep models, which raises the question if this is the case due to the model or the used explanation method. It would be important to discuss this distinction more clearly in the paper."**: The robustness of an explanation method is influenced by the method. For example: smoothgrad is more robust than vanilla gradient or integrated gradient because it averages the attributions. However, the robustness of the method also depends on the underlying model used for explanations. As we demonstrated in section 4.2.3, using the same explanation method but a different model improves the structural similarity of saliency maps. Specifically, the same explanation method produces less robust saliency maps in naturally trained models than in adversarially trained models (with filter smoothing). Stability analysis from Table 1 also confirms how the saliency map stability is tied inherently to the model and reconfirms \citet{ilyas2019adversarial} observation that explanations that are meaningful and faithful to the model's decision-making process cannot be pursued independently from the training of the model, which is a principle central to this work.

---

> > > > ### Comment · Reviewer_iEtN · 2025-04-07
> > > > **Response to rebuttal**
> > > >
> > > > I would like to thank the authors for the revisions to the paper and addressing the requested changes in a detailed manner.
> > > >
> > > > I think all textual changes have clearly improved the flow and structure of the paper.
> > > >
> > > > Regrading the new empirical results, I have some difficulties and reservations.
> > > >
> > > > **Table 2 (Area under perturbation curve for ROAD):** It would be helpful to align Table 2 and Table 1, so that they can be directly compared.  The main insight from this figure is "Results show that naturally trained models are less faithful than adversarially trained models.". While relevant, no clear trends or conclusions regarding the type of adversarial technique are visible. In Fig 4 and Fig 6, I was surprised to see different % percentages of features removed across methods, I would strongly suggest to perform full perturbations (0-100%) and update the ROAD scores accordingly.
> > > >
> > > > Also Thank you for clarifying your use of "Faithfulness". It seems like different definitions of the concept faithfulness are being used, area under perturbation curve for ROAD and the correlation between "attribution sum" with "difference in model prediction". Is there a reason for this? Otherwise I would suggest to streamline your use of Faithfulness and stick to one measure.
> > > >
> > > > Concluding, I would suggest a focused revision with streamlining the writing and focusing your main contributions that are clearly supported by your evidence.

---

> > > > > ### Author Response · Authors · 2025-04-07
> > > > > **Response to reviewer iEtN**
> > > > >
> > > > > We sincerely appreciate the reviewer for their insightful comments. Regarding the ROAD score, the ROAD score in Table 2 are computed for full perturbations (0-100%). In order to visualize the difference between the accuracy drop of different models, we use only 60% for visualization, because as we keep removing more features, all models accuracy drop towards zero. The plots overlap significantly, and its difficult to understand the plot.
> > > > >
> > > > > Thank you suggesting to use a single faithfulness score. We will stick with ROAD and re-compute faithfulness evaluation using ROAD score for plots in 10 and 11 and update the pdf. We will also merge Table 1 and Table 2.
> > > > >
> > > > > On contribution: Our work mainly focuses on the sparsity-stability tradeoff of saliency maps. We included faithfulness evaluation as an additional experiment to evaluate how faithfulness changes with adversarial training and use of smoothing filters. As per the reviewer suggestion, since there are no clear trend or conclusion regarding the faithfulness evaluation, we will revise the paper with focus only on the contributions that are supported by the evidence (sparsity, stability, structural similarity) and do not claim any trend not supported in the experiments (for eg. faithfulness).
> > > > >
> > > > > We will make suitable changes the paper and update the pdf in a few days. Thank you for the insightful suggestions.

---

> > > > > ### Author Response · Authors · 2025-04-10
> > > > > **Updated changes**
> > > > >
> > > > > We thank the reviewer again for their insightful comments. We have made the following changes to the updated paper.
> > > > > - We have softened the claim of faithfulness, focusing on the contributions that are empirically supported. In the abstract, we have removed the claim of ‘faithfulness’ improvement and focused on sparsity-stability tradeoff and comprehensibility evaluation of the qualitative study. In the introduction, we have removed the previous claims of faithfulness and added statements supported by the experiments. We have removed the ‘faithfulness’ improvement claim in conclusion and instead focused on sparsity-stability tradeoff and saliency maps comprehensibility.
> > > > >
> > > > > - We have computed dROAD scores for evaluation similar to sparsity-stability and included the results in Table 1.
> > > > > - We have rewritten Section 4.2.2, discussing dROAD scores and the associated ROAD plots.
> > > > > - We have recomputed the faithfulness evaluation in Sections 4.2.4 and 4.2.5 with the ROAD scores to maintain the consistency of metrics.
> > > > > - We have rewritten the faithfulness claim of Section 4.2.4 with statements that are supported by the results: saliency maps in adversarially trained models with feature map smoothing are more faithful to the underlying model than naturally trained counterparts on CIFAR-10 and ImageNette with Vanilla Gradient and Integrated Gradients.
> > > > > - We have rewritten Section 4.2.4 and Section 4.2.5 by dataset, specifically discussing the lack of faithfulness gain with model robustness in FMNIST.

---

### Review · Reviewer_QSRg · 2025-03-12

**Summary Of Contributions:**

This paper is motivated by the need to improve the reliability and clarity of saliency maps. Typically, input-gradient methods (Vanilla Grad, IG or SmoothGrad) produce explanations that are either noisy (when models are naturally trained) or overly sparse and unstable (when models undergo adversarial training). This trade-off can make the explanations less trustworthy or difficult to interpret.

To address this, the authors propose incorporating feature map smoothing during adversarial training. This approach aims to balance the benefits of sparsity (focusing on the most important features) with stability (robustness to small input perturbations).

Contributions:
* It identifies the trade-off inherent in input-gradient explanation methods, where naturally trained models yield noisy saliency maps and adversarially trained models produce overly sparse and unstable maps.
* It introduces a approach that incorporates feature map smoothing during adversarial training.
* A qualitative user study showing humans preference on attribution maps.

**Audience:**

No

**Claims And Evidence:**

No

**Requested Changes:**

- Section 3.1 provides useful intuition but relies on overly simplified assumptions. It models the network as a single-layer system, which doesn't fully capture the complex interactions and non-linearities of deep architectures. Moreover, the use of L₁ norm bounds may oversimplify how small perturbations affect attribution maps in practice. In more complex networks, interactions among layers or gradient effects might lead to a less direct relationship than the bounds suggest. The reviewer encourages emphasizing this as a limitation of the work.

- In Section 3.2, the authors mainly present qualitative observations without explaining why adversarial training increases sparsity yet decreases stability in saliency maps.

- [for presentation] Tables should be self-descriptive. For example, in Section 3.3, the authors refer to “the introduction of this smoothing block has minimal impact on model accuracy,” which leads the reader to Appendix D. However, Appendix D refers to Table 3, where many abbreviations create confusion regarding what to extract from the table. The reviewer recommends that the authors revise the writing and include the key takeaways from each table in its caption.

- The reviewers encourage the authors to combine related figures (e.g., Figures 4/5/6, 7/8/9, 11/12/13) into one, as the current presentation format hinders readability and direct comparisons across datasets.

- In Section 3.3, while the feature map smoothing approach is intuitive and practical, the authors provide limited theoretical insight into its optimal configuration. For example, the choice of smoothing filter (mean, median, Gaussian, etc.) and its parameters appears largely empirical.

- The paper should reference related work (e.g. [1,2]) that also studies how saliency maps affect “understanding and trusting the underlying model behavior.”

Refs:

- [1] the effectiveness of feature attribution methods and its correlation with automatic evaluation scores
- [2] HIVE: Evaluating the Human Interpretability of Visual Explanations

**Strengths And Weaknesses:**

Strengths:

- The paper presents strong motivations and supports them with empirical observations, highlighting limitations of Vanilla Gradient, Integrated Gradients, and SmoothGrad when used with adversarial training.

Weaknesses:

- The overall writing could be significantly improved for readability (see suggestions).
- The faithfulness results in Figures 4, 5, and 6 are mixed. For example, FMNIST and CIFAR-10 do not consistently demonstrate that saliency maps from robust models are more faithful than those from naturally trained models, which is the key claim. A deeper analysis is needed to clarify whether this inconsistency is due to dataset scale or limitations in the evaluation metrics.
- For the setup of the human study, I believe it should be more quantitative rather than purely qualitative. The better ratings would likely only show the human preference for the authors’ attribution maps that are more comfortable to look at via smoothing. I would strongly suggest evaluation using objective tasks like model debugging[3] or model prediction verification[1,2].

Refs:
- [1] the effectiveness of feature attribution methods and its correlation with automatic evaluation scores
- [2] HIVE: Evaluating the Human Interpretability of Visual Explanations
- [3] debugging tests for model explanations

---

> ### Author Response · Authors · 2025-03-21
> **Response to Reviewer QSRg**
>
> We sincerely appreciate the reviewer for their insightful comments and suggestions. We have uploaded a revised pdf where all the changes are highlighted in blue. Response to suggested changes:
>
> - **Section 3.1 provides useful intuition but relies on overly simplified assumptions**: The primary reason for using a single-layer neural network is to derive closed-form expressions for gradient-based attributions. For deeper networks, the expressions for these attribution methods involve complex compositions of nonlinear transformations and weight matrices, making it infeasible to obtain exact analytical form. The single-layer formulation aligns with prior theoretical work in explainability such as \citet{chalasani2020concise}, which provided closed-form IG expressions under linear model assumptions. However, as the reviewer suggested, this is also a limitation of the study and we have added it to the limitations section.
>
> - **In Section 3.2, the authors mainly present qualitative observations without explaining why adversarial training increases sparsity yet decreases stability in saliency maps**: The increased sparsity in saliency maps due to adversarial training arises from the model learning more selective feature activations, as shown in our feature map visualizations. However, on evaluating the stability of saliency maps, we found the tradeoff with sparsity. This was observed empirically while evaluating the quality of explanations. We hypothesize that since the model's focus is sharpened with selective feature activations, even small natural perturbations in the input could cause large shifts in where it places importance. This leads to instability in the attributions. However, this is just a conjecture and we do not have a theoretical justification yet and will continue looking at this in future works.
>
> - **Tables should be self-descriptive**: We have rewritten all the captions and any mention of the acronym in the main text is also supported by its full form.
>
> - **Readability of related figures**: We have grouped Figures 4,5,6 and 7,8,9 by datasets and rewritten the analysis by dataset such that it improves the readability. We have also placed figures close to the text to help with comparison and reference.
>
> - **In Section 3.3, while the feature map smoothing approach is intuitive and practical, the authors provide limited theoretical insight into its optimal configuration**:  As the reviewer mentions, the inclusion of a smoothing approach is an intuitive method to balance the sparsity-stability tradeoff with the adversarial training. In the paper, we explored several local and non-local smoothing filters, but the choice of the optimal filter and its parameter remains largely empirical and task-dependent. We have added this as our limitation in the limitation section.
>
> - **The paper should reference related work (e.g. [1,2]) that also studies how saliency maps affect “understanding and trusting the underlying model behavior.”**: We agree with the reviewer that qualitative evaluation of explanation methods can also be measured with its application for model debugging or prediction verification. However, in this work, we only focused on comprehensibility of the saliency maps depending on the sparsity level. We have now added the motivation, and also added related works as suggested by the reviewer.
>
> **On faithfulness results:** We have computed dscore as area under perturbation curve as suggested by Reviewer iEtn (Table 2), which helps compare the ROAD plots and make accurate judgements on the faithfulness. We have rewritten the section including this information.

---

> > ### Comment · Reviewer_QSRg · 2025-03-30
> >
> > I would like to thank the authors for the substantial revisions to the paper.
> >
> > I appreciate the effort and believe that the writing has improved significantly.
> >
> > However, I still have concerns regarding the presentation of mixed signals that do not fully support the main claim about the faithfulness of adversarially trained models.

---

> > > ### Author Response · Authors · 2025-03-31
> > > **softening the hard claim**
> > >
> > > We thank the reviewer for raising this concern. We agree that the results for explanation faithfulness across different datasets, attribution methods, and model types vary and do not always support a blanket claim that adversarial training improves faithfulness in a uniform manner. Instead of asserting that adversarially trained models are always more faithful, we can emphasize that feature-map smoothing during adversarial training can lead to more faithful explanations in many cases, depending on the attribution method and dataset used. While we have added limitations of dataset dependent results in the limitation section already, we can also soften the phrasing in our abstract and conclusion to reflect a more balanced and evidence-based contribution.

---

> ### Author Response · Authors · 2025-04-10
> **Changes to the updated paper**
>
> We thank the reviewer again for their insightful suggestions. We have now softened the claim of faithfulness improvement focusing on the contributions that are empirically supported. We have also expanded the limitations section.

---

### Decision · Action_Editor_29dv · 2025-04-20

**Recommendation:** Reject

**Comment:**

The authors updated the paper significantly (9 out of 16 pages significantly changed!) in response to the reviewer comments. Despite the numerous improvements, the updates have not quite resolved the concerns of the reviewers with respect to the clarity of motivation, correctness of claims and conclusions, and sufficient and unambiguous support of these. The updates seem unfortunately also somewhat hasty and rather re-active instead of well thought improvements and upgrades of the paper leaving the paper lacking in overall quality.

In result, I recommend to reject the paper.

The paper nevertheless contains interesting ideas and I recommend the authors to consider re-submitting a major revision, re-thinking the main messages, focusing on communicating these clearly, and providing unambiguous evidence in support of the claims.

**Audience:**

The paper could potentially be of interest for the community. However, at the current state, this is likely to be minimal due to insufficient quality (lack of clarity and convincing evidence for presented claims).

**Claims And Evidence:**

Despite the numerous changes and improvements of the paper in response to reviews, there remain major concerns about the motivation and provided claims and conclusions.

The paper starts with the claim that attribution 'methods struggle to provide explanations that are both visually clear and quantitatively robust' but then instead of focusing on improving these methods, proposes to change the classification models themselves to `yield` better saliency maps. This may make sense under some viewpoints, e.g. `explainability by design`, but if this is the case or not should made clear from the outset of the paper to help the reader understand the setting.

Despite the updated wording, the conclusions of the paper remain ambiguous. There is a rather large discussion around the trade-offs between sparsity, stability, faithfulness and accuracy. The final conclusions `Incorporating local feature-map smoothing during adversarial training enhances stability and comprehensibility without sacrificing sparsity.` are, however, not sufficiently corroborated by the evidence in Table 1 which shows rather variable performance in the stability metrics across the datasets, attribution methods and local smoothing approaches.

Two out of the three reviewers found the updates of the paper and answers of the authors not sufficient to dissolve their concerns about the presented claims.

**Resubmission Of Major Revision:**

The authors may consider submitting a major revision at a later time.